# SPARSE IMAGINATION
# FOR EFFICIENT VISUAL WORLD MODEL PLANNING

**Junha Chun**[2*]   **Youngjoon Jeong**[1*]   **Taesup Kim**[1†]
[1]Graduate School of Data Science, Seoul National University
[2]Department of Electrical and Computer Engineering, Seoul National University

## ABSTRACT

World model based planning has significantly improved decision-making in complex environments by enabling agents to simulate future states and make informed choices. This computational burden is particularly restrictive in robotics, where resources are severely constrained. To address this limitation, we propose a *Sparse Imagination for Efficient Visual World Model Planning*, which enhances computational efficiency by reducing the number of tokens processed during forward prediction. Our method leverages a sparsely trained vision-based world model based on transformers with randomized grouped attention strategy, allowing the model to flexibly adjust the number of tokens processed based on the computational resource. By enabling sparse imagination during latent rollout, our approach significantly accelerates planning while maintaining high control fidelity. Experimental results demonstrate that sparse imagination preserves task performance while dramatically improving inference efficiency. This general technique for visual planning is applicable from simple test-time trajectory optimization to complex real-world tasks with the latest VLAs, enabling the deployment of world models in real-time scenarios.

## 1 INTRODUCTION

By "imagining" future trajectories in a learned world model of the environment, agents can perform sophisticated decision making without trial-and-error in the real world (Ha & Schmidhuber, 2018; Hafner et al., 2019a;b). Recent world model advancements for visual control tasks, which necessitate reasoning over high-dimensional image observations, increasingly leverage high-level visual features derived from self-supervised vision models (Caron et al., 2021; Oquab et al., 2023; Nair et al., 2022; Xiao et al., 2022). For example, DINO-WM (Zhou et al., 2024) introduces a world model that predicts future Vision Transformer (ViT) patch tokens (DINO features) instead of pixels or single vector representation, enabling zero-shot planning for diverse tasks. These approaches retain rich spatial tokens, achieving strong generalization on complex control tasks. However, using numerous visual tokens for planning incurs a prohibitive quadratic computational cost, as it requires many simulated rollouts. This raises a crucial question:

> *Can we retain the advantages of detailed visual world models while enhancing computational efficiency for planning?*

In this paper, we address the above question by introducing the concept of *sparse imagination*, accelerating world model inference by deliberately using only a sparse subset of visual tokens during forward prediction. Our key insight is that ViT-based representations are redundant (Raghu et al., 2021; Pan et al., 2021; Chen et al., 2022; Kim et al., 2024; Siméoni et al., 2025); thus, not all patch tokens are necessary for planning.

To achieve this, we propose a random dropout-based token selection mechanism for the world model during inference that dynamically encodes and predicts only a subset of patch tokens. Furthermore, we train the world model using randomized grouped attention to accommodate dynamic token selection. The resulting system performs planning on the subset of latent patch tokens, reducing

---

*These authors contributed equally.
†Corresponding author.

computational cost while maintaining competitive performance. The primary contributions of this work are as follows:

- We introduce sparse imagination, a simple yet effective method that enables efficient visual world model planning by leveraging random patch feature dropout during inference.

- We demonstrate that our approach is a general technique applicable from simple test-time trajectory optimization to complex real-world tasks with VLAs. It achieves substantial planning speedups while maintaining high performance, validated on benchmarks like LIBERO-10 (Liu et al., 2023a), Meta-World (Yu et al., 2021), and on real-world robotic tasks.

- Our comparative analysis shows that simple random sampling achieves competitive or superior performance to sophisticated token selection methods due to a fundamental "blind spot" problem we identify, where static importance metrics fail in dynamic planning, making the unbiased coverage of random sampling a more robust strategy.

## 2 RELATED WORK

**World Models for Control.** Early world models transitioned from computationally heavy pixel-level models (Finn et al., 2016; Ebert et al., 2018) to latent dynamics models like PlaNet (Hafner et al., 2019b) and Dreamer (Hafner et al., 2019a; 2020; 2023), which achieved impressive results by leveraging planning and latent imagination. While most existing latent world models compress entire images into relatively low-dimensional vectors, these compact representations often lose the fine-grained spatial information necessary for high-precision manipulation tasks (Zhou et al., 2024; Tsagkas et al., 2025).

**Visual Representations for Decision Making.** A recent trend in decision making leverages powerful pre-trained visual encoders for rich state representations (Nair et al., 2022; Xiao et al., 2022). While effective for representation learning, methods using a single image embedding (e.g., CLS token) from encoders like ResNet or ViT often lose crucial fine-grained spatial details, hindering performance on tasks requiring precise spatial understanding, such as manipulation (Tsagkas et al., 2025; Zhou et al., 2024). To address this, approaches using spatial features like ViT patch embeddings for decision making have been developed, exemplified by DINO-WM (Zhou et al., 2024) which uses a set of DINO patch tokens as the state. Building on this paradigm, our work utilizes pre-trained ViT patch features for their spatial granularity. Our innovation is an inference-time token dropping mechanism that significantly enhances computational efficiency during imagination.

**Vision Transformers and Token Efficiency.** The quadratic scaling of ViT self-attention poses a significant computational challenge, particularly for sequential models like world models (Dosovitskiy et al., 2020). Improving inference efficiency via token reduction is thus a key vision research area. Existing methods employ diverse strategies such as learned/attention-based selection (Rao et al., 2021; Luo et al., 2024; Yin et al., 2022; Liang et al., 2021; Wang et al., 2022; Zhang et al., 2024b), merging (Bolya et al., 2022; Feng & Zhang, 2023; Haurum et al., 2024), and training-time dropout (Liu et al., 2023b). These demonstrate efficiency gains from ViT representation redundancy (Raghu et al., 2021; Pan et al., 2021; Chen et al., 2022; Kim et al., 2024). In contrast, we present a patch-dropping method specifically for inference-time world model planning. Our approach uniquely applies dropout only during the imagination phase, enabling seamless integration with any transformer world model.

## 3 METHODOLOGY

### 3.1 WORLD MODEL

Our world model is composed of two key modules: (1) a pre-trained image encoder $g_\phi$ (e.g., DINO) generating visual latent patch tokens from observations, and (2) a transformer-based world model $f_\theta$ predicting future latent tokens based on historical observations and actions. The image encoder remains fixed during training to ensure stable visual representations, while the transformer-based world model learns temporal dynamics. Structured as a causal transformer decoder, the world model

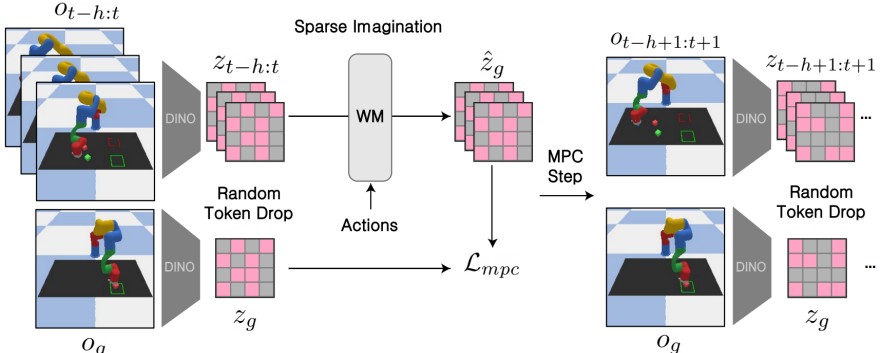

Figure 1: **Sparse Imagination.** Sparse imagination accelerates planning by performing model predictive control (MPC) rollouts on a random subset of visual tokens. A new dropout pattern is dynamically sampled at each MPC iteration, and both predictions and optimization for CEM are computed using only the selected patches to improve efficiency and robustness.

allows each token to attend only to tokens from prior time steps, enabling efficient sequence modeling.

Specifically, given an image observation $o_t \in \mathbb{R}^{H \times W \times 3}$ at time $t$, the image encoder $g_\phi$ extracts visual features $z_t \in \mathbb{R}^{H_p \times W_p \times D}$, where the number of visual tokens is $N = H_p \times W_p$. The world model $f_\theta$ predicts future features $z_{t+1}$ using a temporal context of visual tokens and actions: $z_{t+1} = f_\theta(z_{t-h:t}, a_{t-h:t})$, with a history length of $h + 1$. Our model predicts dynamics within the pre-trained DINO feature space. Actions and proprioceptive information, when available, are linearly projected and concatenated to visual tokens along the feature dimension. The world model training is guided by minimizing the mean squared error (MSE) prediction loss:

$$\mathcal{L}_{\text{wm}} = \frac{1}{N} \sum_{i=1}^{N} ||\hat{z}_{t+1,i} - z_{t+1,i}||^2, \tag{1}$$

where $i$ and $N$ represent token index and the total visual tokens per frame, respectively. An optional decoder, detached from gradient updates, can be trained separately for visualization purposes.

To enhance generalization and robustness when only sparse subsets of tokens are provided as inputs, we introduce a randomized grouping strategy during training as depicted in Fig. 2. Specifically, visual tokens from each frame are randomly partitioned into two groups. This approach ensures different subset patterns during training, promoting adaptability across different levels of sparsity. To implement this, we apply attention masks within the transformer layers, restricting token interactions exclusively to those within the same spatial group while maintaining temporal consistency. By enforcing this structured token separation, our model learns to process arbitrary subsets of visual tokens while optimizing for the same prediction loss, ultimately improving its ability to generalize across varying input conditions.

## 3.2 SPARSE IMAGINATION FOR MPC

We use MPC for planning, and for each step, candidate action sequences $\{a_{t:t+H-1}\}$ are sampled from a distribution (CEM) or pre-trained policy. The trained world model $f_\theta$ then predicts future states by iteratively rolling out these action sequences from the current state. In a goal-conditioned setting, we evaluate candidate sequences using the MSE loss between the predicted and goal state features:

$$\mathcal{L}_{\text{mpc}} = ||\hat{z}_{t+H} - z_g||^2. \tag{2}$$

When using CEM for test-time trajectory optimization, the distribution of candidate actions is subsequently updated based on the top-performing sequences. This iterative optimization continues for a predefined iteration step $M$, after which the optimized action sequence is executed in the environment.

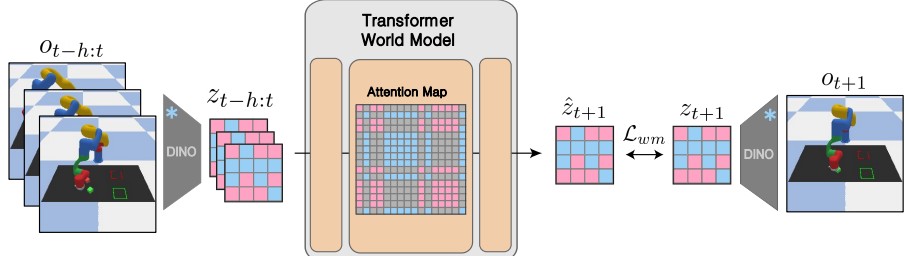

Figure 2: **Randomized Grouped Attention Strategy.** Our randomized grouping strategy used during training to generalize to arbitrary token subsets. Visual tokens are randomly partitioned into two groups, and attention is masked to occur only within the same spatial group. This trains the model to process sparse inputs effectively, and its necessity is shown in ablations.

MPC is computationally intensive, as world model rollout occurs $K \times M \times H$ per every planning step, and this cost scales quadratically with the number of visual tokens (the number of candidate sequences is $K$ and planning horizon is $H$), making real-time execution increasingly challenging. Since attention layers in the world model are the primary source of this computational burden, reducing the number of tokens processed is crucial for improving planning efficiency.

We address this problem through *sparse imagination*. Given that our model is explicitly trained to operate on arbitrary token subsets, we implement random token dropout during planning as shown in Fig. 1. At each planning step, we randomly generate a token dropout mask characterized by a dropout fraction $p \in [0, 1)$ of the user's choice, denoting the proportion of tokens to discard, keeping randomly sampled $(1 - p)N$ tokens. Sparse rollout imagination can effectively reduce redundant computations and improve planning efficiency. Planning a single step might fall to wrong direction when dropping key features but has minimal impact on task performance since it can recover from failures by re-sampling at each planning iteration. The drop ratio $p$ serves as a controllable parameter balancing computational speed and task accuracy. We empirically evaluate and analyze this trade-off in the following Sections.

## 4 EXPERIMENTS

### 4.1 IMPLEMENTATION DETAILS

**Environments and Datasets.** We evaluate our method on eight simulated environment and two real-world tasks using LeRobot (Cadene et al., 2024) (See Fig. 3). Detailed descriptions of the environments, datasets, and model configurations are provided in the Appendix A.

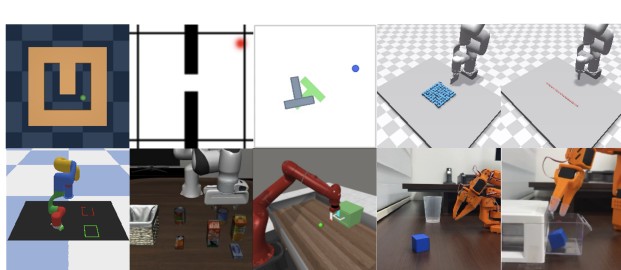

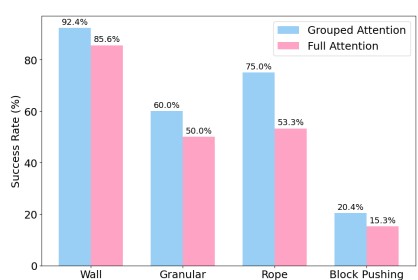

Figure 3: **Environments.** We assess the world model across eight simulation environments and two real-world tasks, arranged as follows (left-to-right, top-to-bottom): Pointmaze, Wall, PushT, Granular, Rope, Block Pushing, LIBERO, Meta-World and LeRobot tasks (PickPlace & Drawer).

Figure 4: **Contribution of grouped attention during pretraining to planning.** We compare the average planning success rates achieved by grouped attention and full attention under various drop ratios, highlighting the benefit of grouped attention.

Table 1: Performance results (Mean Success Rate, %) across different environments with varying drop ratios.

|  | Pointmaze (60 evals) | Wall (60 evals) | PushT (60 evals) | Granular (20 evals) | Rope (30 evals) | Block Pushing (50 evals) | Avg. |
|---|---|---|---|---|---|---|---|
| Full | 98.3 | 91.7 | 75.0 | 75.0 | 63.3 | 22.0 | 69.8 |
| CLS | 96.7 | 85.0 | 43.3 | 20.0 | 36.7 | 16.0 | 50.7 |
| Drop (Ratio) | | | | | | | |
| 10% | 91.7 | 93.3 | **78.3** | 80.0 | 73.3 | **28.0** | 74.1 |
| 20% | 90.0 | **95.0** | 70.0 | 80.0 | 66.7 | 16.0 | 69.6 |
| 30% | 98.3 | 93.3 | 61.7 | **85.0** | 70.0 | 18.0 | 71.0 |
| 40% | 96.7 | 93.3 | 56.7 | 70.0 | **76.7** | 18.0 | 68.6 |
| 50% | **100.0** | **95.0** | 70.0 | 60.0 | 73.3 | 20.0 | 69.7 |
| 60% | 83.3 | 91.7 | 46.7 | 70.0 | 70.0 | 20.0 | 63.6 |
| 70% | 70.0 | 93.3 | 28.3 | 55.0 | 53.3 | 20.0 | 53.3 |
| 80% | 73.3 | 90.0 | 21.7 | 50.0 | 66.7 | 24.0 | 54.3 |
| 90% | 71.7 | 86.7 | 20.0 | 40.0 | 50.0 | 12.0 | 46.7 |

In each environment, the goal is reaching a target observation randomly sampled as a desired goal state, starting from initial positions. For the LIBERO environment, we conducted experiments on LIBERO-10, a suite of long-horizon tasks, to demonstrate the effectiveness of our planning acceleration. For the real world, the task is to pick the blue block and place it in the cup.

**Baselines.** We evaluate our approach across several visual control domains and compare it against key baselines. The primary baselines include: (1) a **Full-Patch** world model, DINO-WM (Zhou et al., 2024) based on DINO (Caron et al., 2021), which shares the same architecture as ours but utilizes all patches during planning (i.e., $p = 0$) and applies positional encoding to each patch—representing a high-fidelity yet computationally expensive setting; and (2) a **CLS-Token** world model, which employs only the global CLS token from the DINO encoder as the visual representation, thereby offering a single-vector representation baseline. These two baselines are based on the approach presented in DINO-WM.

**Planning Setup.** To evaluate our framework across diverse scenarios, we utilize three distinct planning schemes:

- **MPC-CEM:** For Pointmaze, Wall, PushT, Block Pushing, we use MPC-CEM. This involves replanning at each step in a receding-horizon manner.

- **CEM (Open-loop):** For computationally intensive deformable object simulations (Granular, Rope), we employ a standard open-loop CEM approach. This optimizes a single, fixed sequence of actions without replanning to manage the high processing cost.

- **Policy-Guided Planning:** For complex, long-horizon tasks (LIBERO, Meta-World, LeRobot), where random action sampling is inefficient, we guide the planner by sampling $K$ candidate action sequences from a pre-trained policy such as Vision-Language-Action (VLA) models. Our planner then evaluates these candidates.

Our contribution highlights significant efficiency gains while preserving **Full-Patch (DINO-WM)** baseline performance, demonstrating our sparse imagination as an effective, computationally efficient replacement, rather than focusing on absolute performance maximization. Therefore, for fair comparison, most of hyperparameters are adopted from the DINO-WM baseline, as detailed in the Appendix A.3 and B.1.

## 4.2 Test-Time Trajectory Optimization

Our sparse imagination framework is first evaluated on test-time trajectory optimization using MPC-CEM or CEM. Moderate token dropout (10-50%) maintains task performance relative to the **Full-Patch** baseline while substantially reducing planning time. For example, in the PushT environment, a 50% dropout ratio cuts the planning time per iteration from 173s to 82s (a 52.6% reduction) without sacrificing performance (Table 1, 2).

Table 2: Planning time and change compared to Full baseline for different environments at various drop ratios.

| | Pointmaze | | Wall | | PushT | | Block Pushing | | Avg. | |
|---|---|---|---|---|---|---|---|---|---|---|
| | Planning Time (s/iter) | Change (%) | Planning Time (s/iter) | Change (%) | Planning Time (s/iter) | Change (%) | Planning Time (s/iter) | Change (%) | Planning Time (s/iter) | Change (%) |
| Full | 184 | 0.0 | 79 | 0.0 | 173 | 0.0 | 297 | 0.0 | 183.3 | 0.0 |
| CLS | 49 | -73.4 | 40 | -49.4 | 32 | -81.5 | 163 | -45.1 | 71.0 | -61.3 |
| Drop (Ratio) | | | | | | | | | | |
| 10% | 165 | -10.3 | 73 | -7.6 | 149 | -13.9 | 278 | -6.4 | 166.3 | -9.3 |
| 20% | 141 | -23.4 | 69 | -12.7 | 131 | -24.3 | 259 | -12.8 | 150.0 | -18.1 |
| 30% | 126 | -31.5 | 65 | -17.7 | 114 | -34.1 | 240 | -19.2 | 136.3 | -25.6 |
| 40% | 106 | -42.4 | 62 | -21.5 | 97 | -43.9 | 214 | -27.9 | 119.8 | -34.7 |
| 50% | 93 | -49.5 | 53 | -32.9 | 82 | -52.6 | 208 | -30.0 | 109.0 | -40.5 |
| 60% | 80 | -56.5 | 50 | -36.7 | 69 | -60.1 | 200 | -32.7 | 99.8 | -45.6 |
| 70% | 69 | -62.5 | 46 | -41.8 | 59 | -65.9 | 184 | -38.0 | 89.5 | -51.2 |
| 80% | 56 | -69.6 | 46 | -41.8 | 49 | -71.7 | 175 | -41.1 | 81.5 | -55.5 |
| 90% | 48 | -73.9 | 42 | -46.8 | 38 | -78.0 | 167 | -43.8 | 73.8 | -59.8 |

Table 3: Performance results and planning time across difficulties in Meta-World simulation.

| | All (%) | Easy (%) | Medium (%) | Hard (%) | Very Hard (%) | Planning Time (s/episode) |
|---|---|---|---|---|---|---|
| VLA-only | 41.87 | 60.95 | 20.61 | 17.78 | 10.67 | 1.95 |
| Full | 48.80 | 66.90 | 29.09 | 28.89 | 14.67 | 3.63 |
| Drop 50% | 47.73 | 65.24 | 27.27 | 33.33 | 12.00 | 2.37 |

This efficiency gain is especially critical in complex environments. Although the **CLS-Token** offers faster inference, its dramatic performance drop in these scenarios renders it impractical. For instance, in spatially demanding scenes like Granular and Rope, our approach achieves up to 85.0% and 76.7% success respectively, whereas the **CLS-Token** fails (20.0%, 36.7%) by losing critical spatial information. Similarly, on tasks like PushT, our method with 50% dropout (70.0%) surpasses **CLS-Token** (43.3%) baselines. Refer to Appendix B.8 for more detailed and fine-grained analysis.

## 4.3 Towards Complex and Real-World Tasks: VLA-Guided Planning

To tackle complex and long-horizon tasks, we integrate our planning framework with the latest VLA model such as SmolVLA (Shukor et al., 2025). We evaluate this approach on the LIBERO-10 and Meta-World simulation and the real-world environments using SO-101 robot arm, focusing on two tasks: **PickPlace** (picking up a block and placing it into a cup) and **Drawer** (picking up a block, placing it into a drawer, and then closing the drawer). We find that guiding the search with a pretrained policy is crucial in these demanding settings. For the real-world LeRobot experiments on **PickPlace** and **Drawer**, we used a manually selected goal image and proprioceptive state from the dataset for planning (See Appendix A.8 for details).

**Simulation Results.** The SmolVLA policy used for simulation experiments is trained to predict action chunks spanning 50 steps and at each decision step we sample 50 candidate action chunks from the policy for LIBERO-10 and 10 for Meta-World to perform planning-based evaluation. In the original work of SmolVLA, only a single action is performed in every policy inference. Since we find it impractical to deploy single-step policies in real world, we execute all actions in the action chunk despite lower absolute performance. In LIBERO-10 evaluation, we test our method in the evaluation benchmark setting with total of 100 trials. Our method (50% token drop, ★ in the Fig. 5) boosted the base VLA's success rate from 29% to 33%. This performance matches the computationally expensive **Full-Patch** planner but completes episodes nearly twice as fast (29.7s vs. 53.4s per episode). Although it is slower than the base policy due to additional planning overhead, this can be efficiently handled in real-world deployment which will be stated in following section. In Meta-World evaluation, we test on 50 tasks with 15 trials each. As shown in Table 3, our method also show comparable performance to the **Full-Patch** planner with lower computation overhead.

**Real-world LeRobot Results.** In our real-world LeRobot evaluation on **PickPlace** and **Drawer**, we tested by fixing the cup's position or drawer pose and placing the block in initial locations unseen during training for both tasks. We sample 10 distinct candidate chunks from the policy at each decision step instead of 50 for real-world deployment. On a physical robot, sparse imagination

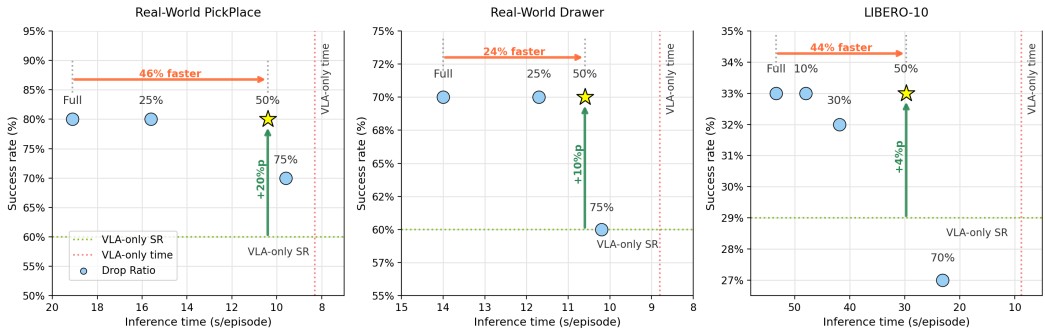

Figure 5: **Trade-off between Inference Time and Performance in LeRobot and LIBERO-10.** We show performance of SmolVLA with sparse imagination planning with a particular drop ratio in Left: Real-World PickPlace; Center: Real-World Drawer; Right: LIBERO-10; where (★) represent the operating point we highlight (50% drop), which provides a practical choice between speed and reliability.

with a 50% drop had an even greater impact, significantly increasing the success rate from 60% (VLA-only) to 80% and 70% on **PickPlace** and **Drawer**, respectively. Again, this matched the **Full-Patch** planner's success rate while reducing planner latency from 19.1s to 10.4s on **PickPlace** and from 14.0s to 10.6s on **Drawer** per episode. By leveraging asynchronous inference, this major performance gain across both tasks is achieved with minimal overhead compared to the VLA-only baselines (8.3s and 8.8s), proving our method is a practical solution for real-time robotic planning scenarios.

## 4.4 TOKEN REDUCTION METHODS

We compare our random sampling against a broad set of token-selection strategies, grouped into four families as summarized in Table 4: (i) **Random Sampling** (Random, Fixed, LHS), (ii) **Learning-Based Pruning** (LTRP), (iii) **Attention-Based Pruning** (Attention-Encoder, STAR, Attention-WM), and (iv) **Cluster & Merging** (ATC). All methods share the same world model and planning hyperparameters, and retain the same number of tokens for a fair comparison.

**Random Sampling.** The Random baseline samples a new subset of patch tokens independently at every planning iteration, while Fixed draws a single subset once for each rollout and reuses it for all rollouts within the episode. We additionally include LHS (Latin Hypercube Sampling; (McKay et al., 2000)), which performs stratified sampling over the 2D patch grid so that retained tokens are approximately uniformly distributed across the image.

**Learning-Based Pruning.** LTRP (Learning to Rank Patches; (Luo et al., 2024)) trains a ranking network on top of a pre-trained MAE (He et al., 2022) to predict patch importance based on reconstruction sensitivity; at test time, the top-$K$ tokens according to this learned score are kept.

**Attention-Based Pruning.** Attention-Encoder and STAR (Zhang et al., 2024b) derive token importance from the self-attention maps of a pre-trained DINO encoder, while Attention-WM uses attention maps from our ViT-based world model itself. In all three cases, we rank tokens by their attention-based scores and retain the top-$K$ as the "important" subset.

**Cluster & Merging.** ATC (Haurum et al., 2024) clusters spatially similar patch tokens (using DINO features) and replaces each cluster with its average, effectively merging redundant patches. For goal-conditioned planning, both the current observation and goal features are clustered, and the distance between predicted futures and goal is computed after aligning clusters via Hungarian matching. A full description of each method and its training configuration is provided in Appendix A.6.

Despite their complexity, empirical results across various datasets consistently show that none of these methods significantly outperform our simple random sampling (Table 4). Fig. 6 visualizes the distinct dropout patterns for each approach.

Table 4: The average planning success rate (%) achieved by various token dropout and merging methods under different drop ratios. Our results suggest that no method achieves significantly superior performance compared to the Random sampling. "Fixed" is excluded for Granular and Rope because they only use CEM, making "Random" and "Fixed" strategies equivalent. For the "Fixed" variant in Granular and Rope tasks, the Avg. is computed by reusing the drop ratios of the corresponding "Random" setting.

| Type | Methods | Pointmaze | Wall | PushT | Granular | Rope | Block Pushing | Avg. |
|---|---|---|---|---|---|---|---|---|
| Random Sampling | Random | 85.8 | 93.8 | **50.4** | **82.5** | **67.5** | 20.5 | **66.7** |
| | Fixed | **87.1** | 85.0 | 47.4 | - | - | 16.5 | 64.3 |
| | LHS (McKay et al., 2000) | 84.2 | 92.5 | 49.6 | **82.5** | 64.2 | **21.0** | 65.7 |
| Learning-Based Pruning | LTRP (Luo et al., 2024) | 79.2 | 94.6 | 40.9 | 75.0 | 50.8 | 16.5 | 59.5 |
| Attention-Based Pruning | Attention-Encoder | 79.2 | **97.1** | 43.9 | 70.0 | **67.5** | 20.5 | 63.0 |
| | STAR (Zhang et al., 2024b) | 82.9 | 85.8 | 46.9 | 72.5 | 63.3 | 17.0 | 61.4 |
| | Attention-WM | 80.0 | 93.8 | 44.3 | **82.5** | 66.7 | 16.5 | 64.0 |
| Cluster & Merging | ATC (Haurum et al., 2024) | 78.3 | 81.2 | 39.1 | 15.0 | 25.0 | 11.5 | 41.7 |

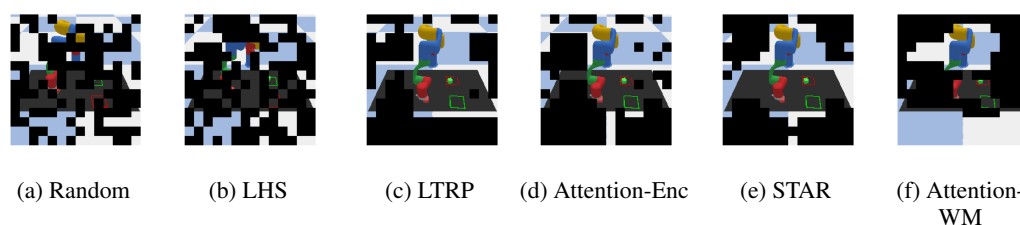

| (a) Random | (b) LHS | (c) LTRP | (d) Attention-Enc | (e) STAR | (f) Attention-WM |
|---|---|---|---|---|---|

Figure 6: **Visualization of token dropout patterns for different token reduction methods.** Patterns at 50% dropout ratio are shown from left to right for (a) Random, (b) LHS, (c) LTRP, (d) Attention-Encoder, (e) STAR, and (f) Attention-WM.

## 5 ABLATION AND ANALYSIS

### 5.1 ARCHITECTURAL GENERALIZATION

To demonstrate broader applicability, we evaluated our sparse imagination framework with MoCo-v3 ViT-S (Chen et al., 2021), MAE ViT-B (He et al., 2022), and DINOv3 ViT-S (Siméoni et al., 2025) in the PointMaze environment (60 episodes). With a 50% token drop, our method matched the MoCo-v3 **Full-Patch** baseline's 96.7% success and the DINOv3 **Full-Patch** baseline's 98.3% success, while outperforming the MAE **Full-Patch** baseline (75.0% vs. 68.3%). These results confirm that sparse imagination generalizes effectively across diverse vision encoders.

### 5.2 ROBUSTNESS TO TOKEN SPARSITY DURING PLANNING

Training the world model with our randomized grouped attention is crucial for leveraging these sparse token subsets. This strategy significantly reduces prediction errors on token subsets compared to a standard full-attention model (e.g., 0.016 vs. 0.036 of normalized $L_2$ error for 50% drop), which directly translates to improved planning performance (Fig. 4). We also find that the CEM planner is robust enough to handle the small remaining prediction noise, based on the experimental results, which is consistent with prior studies (De Boer et al., 2005; Hu & Hu, 2009; Lale et al., 2024). Detailed experiments evaluating CEM's performance with visual token errors in our framework are in Appendix B.3.

### 5.3 INFORMATION SUFFICIENCY OF SPARSE TOKEN SUBSETS

Our analysis confirms that a sparse subset of tokens is sufficient for effective planning. We found that even a small, random fraction of tokens retains substantial ground-truth state information, often more than the global CLS token, as measured by both an information-theoretic analysis using normalized

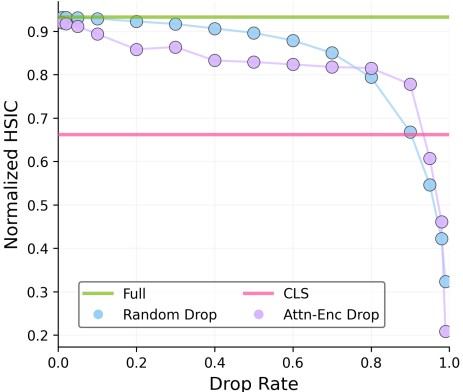
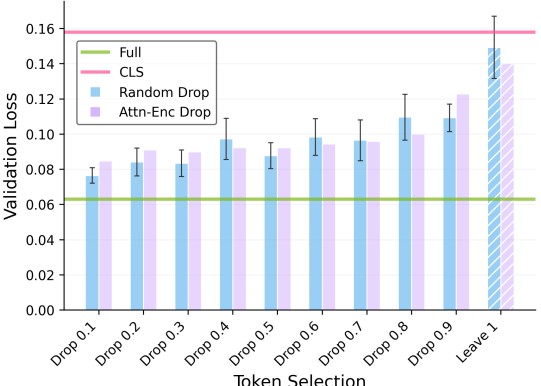

Figure 7: **nHSIC between visual tokens and environment states.** nHSIC scores with respect to dropout rate for random dropout compared to Full, CLS, and Attention-Encoder dropout in Granular dataset. High HSIC scores are maintained even under substantial dropout, indicating minimal loss of critical state information.

Figure 8: **Attentive probing performance.** Validation losses for state prediction with attentive probing compared to Attention-Encoder baseline in Granular dataset. Surprisingly, a single random token shows predictive accuracy comparable (albeit with variance) to the CLS token, indicating state information is distributed across tokens.

Hilbert-Schmidt Independence Criterion (Gretton et al., 2005), (Fig. 7) and an attentive probing module (Bardes et al., 2024) (Fig. 8). Methodological details are deferred to the Appendix A.4 and A.5.

## 5.4 EXPLAINING THE COUNTER-INTUITIVE SUCCESS OF RANDOM SAMPLING

Counter-intuitively, sophisticated importance-based sampling methods failed to show a clear advantage over simple random sampling, often underperforming it (Table 4 vs. Fig. 6).

**The "Blind Spot" Problem of Importance-Based Sampling.** We trace this phenomenon to a fundamental "blind spot" problem. Methods based on static importance metrics, by design, learn to ignore regions of the scene deemed unimportant from the static initial and goal images. This proves fatal in dynamic tasks. If an object of interest enters a blind spot, the predicted visual features for different candidate actions become indistinguishable to the planner. This causes the candidate selection process to collapse, as it cannot identify the best action based on visual progress. The failure is starkly demonstrated in a vision-only planning experiment (Wall, 60 episodes, 50% dropout, proprioception distance is not used): the Attention-Encoder method's success rate collapsed to 21.7%, whereas random sampling, which avoids such blind spots, remained robust at 58.3%. A control experiment using only proprioception confirmed vision's essential role, achieving just 16.7% success. We provide more detailed analysis in Appendix B.9, and visualization example of this blind-spot failure in Figs. 9 and 20.

**Robustness from Unbiased Coverage and Information Redundancy.** The success of random sampling lies in its unbiased coverage, which is effective precisely because task-relevant information is highly redundant and distributed. We quantify its lack of bias using "cluster retention entropy", a metric showing random sampling achieves significantly more uniform coverage than Attention-Encoder (2.37 vs. 1.91 bits, higher is more uniform). The viability of this unbiased strategy is confirmed by a critical finding: planning with only the tokens deemed least important by the Attention-Encoder still yielded a strong 63.7% success rate (compared to 79.2% with the most important tokens; PointMaze, 60 episodes, 50% drop). This proves that a random subset is almost certain to contain sufficient information for planning. This finding of distributed information is further validated by our information-theoretic analyses in Section 5.3. More experimental details are presented in Appendix A.7.

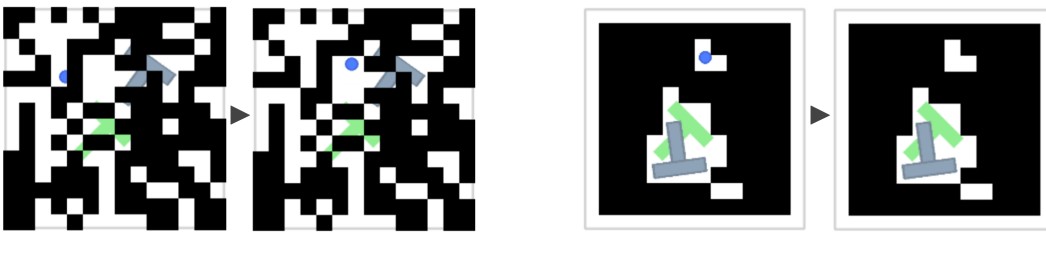

(a) Random                   (b) Importance-Based (Attention-Encoder)

Figure 9: **Visualization Example of the Blind Spot Problem.** Both images show 60% token dropout masks in PushT: (a) **Random** sampling and (b) an **importance-based** method (Attention-Encoder). In the importance-based case, the blue ball's path toward any direction to solve the task is almost entirely covered by masked patches, creating a persistent blind spots where the world model cannot observe the object's movement and thus fails to plan correctly. Random sampling, by contrast, spreads retained tokens more uniformly, making such task-relevant regions less likely to be systematically ignored.

Our analyses resolve the paradox of random sampling's success: its unbiased coverage prevents the fatal "blind spots" that plague static, importance-based methods. This strategy is effective precisely because, as our experiments show, task-relevant information is broadly distributed across the image, ensuring even a random subset likely contains sufficient information for robust planning.

## 6   CONCLUSION

In this paper, we introduced sparse imagination, a novel approach for computationally efficient planning in transformer-based world models. By processing only a subset of visual tokens via random dropout, our method significantly reduces computational complexity while maintaining high task performance across tasks ranging from simple test-time optimizations to complex real-world applications with the latest VLAs. Our comparative analysis showed that various sophisticated sampling methods failed to outperform simple random sampling, and our investigation into the "blind spot" problem suggests this is due to a fundamental mismatch between their static importance metrics and the dynamic nature of planning.

While more advanced, dynamic-aware strategies might exist, they must justify their inherent complexity and computational overhead. For practical applications under resource constraints, we argue that a simple, near-zero-overhead strategy like random dropout provides a powerful and effective baseline, whose practicality may ultimately outweigh the marginal gains of more complex alternatives. The efficiency gains from this approach can then be reallocated to widen the action search or process a longer history of observations. This work establishes random dropout as a simple, robust, and highly practical baseline for deploying visual world models in real-time robotic applications.

**Limitations.** Our work has two primary limitations. First, the efficacy of our approach is inherently dependent on the quality of the underlying visual representations from the pre-trained encoder. Second, while we demonstrate that a moderate dropout ratio is a robust default in our benchmarks, we do not claim that any particular fixed ratio is universally optimal, and the ideal sparsity level is likely task-dependent. Although we explore a preliminary uncertainty-aware adaptive mechanism in Appendix B.6, developing a more sophisticated method to adaptively adjust this ratio remains a promising avenue for future research.

**Reproducibility Statement.** We strongly encourage the reproducibility of our work. To facilitate this, we provide the maximum possible level of detail regarding our experimental setup in the Appendix A. This includes a comprehensive description of the datasets used, the full training procedures, model architectures, and detailed hyperparameter settings.

**Acknowledgements.** This work was supported by the National Research Foundation of Korea (NRF) grant funded by the Korea government (MSIT) (No. RS-2024-00345809, Research on AI Robustness Against Distribution Shift in Real-World Scenarios; and No. RS-2023-00222663, Center for Optimizing Hyperscale AI Models and Platforms).

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

APPENDIX

# A IMPLEMENTATION DETAILS

## A.1 DATASETS AND ENVIRONMENT DETAILS

We elaborate various datasets and environments where our method is experimented on (Table 5). Offline trajectory datasets of Pointmaze, Wall, PushT, Granular and Rope as well as some of dataset-specific configurations are adopted from those provided by Zhou et al. (2024)[1]. Configurations specific to dataset include history trajectory length for the world model and frame skip length, which is an interval of frames to and from the world model, reducing redundancy between consecutive inputs. We also differ training epoch due to difference of dataset size and complexity.

Table 5: Dataset details and dataset-specific configurations. Note that history length, frame skip, and train epochs are for training world model.

| Dataset | Size | # Frames | History len | Frame skip | Train epoch |
|---|---|---|---|---|---|
| Pointmaze | 2000 | 100 | 3 | 5 | 10 |
| Wall | 1920 | 50 | 3 | 5 | 100 |
| PushT | 20000 | 100∼300 | 3 | 5 | 1 |
| Rope | 1000 | 20 | 1 | 1 | 100 |
| Granular | 1000 | 20 | 1 | 1 | 100 |
| Block Pushing | 11000 | 200 | 3 | 5 | 4 |
| LIBERO | 1693 | 75∼505 | 3 | 5 | 10 |
| Meta-World | 2500 | 14∼500 | 3 | 5 | 10 |
| LeRobot | 100 | 208∼459 | 3 | 5 | 50 |

**Pointmaze.** A navigation task introduced in D4RL (Fu et al., 2020) where an agent interacts with given 2D maze environment. There are 2000 trajectories with length of 100 created with random actions. Since we use task-agnostic world model with goal-conditioned planning, goal point is not visualized in each frame.

**Wall.** A visually simplistic 2D navigation task where an agent needs to pass a door to a separated room. There are 1920 trajectories with length of 50 created with random actions.

**PushT.** A 2D control task introduced in Chi et al. (2024) where an agent needs to manipulate T-shaped block to a designated location. Dataset is generated by rollouting expert demonstration action trajectory with additional action noise in the environment. This allows the world model to learn various dynamics.

**Granular and Rope.** Fine-grained manipulation tasks involving deformable objects introduced in Zhang et al. (2024a). Both datasets have 1000 trajectories with length of 20 created with random actions.

**Block Pushing.** Robotics control task requiring precise spatial reasoning to position blocks correctly. We generate 11000 trajectories following PushT generation process.

**LIBERO.** The LIBERO benchmark is a multitask robot learning environment featuring a diverse set of objects and tasks. For training, we utilized only the fixed-view images from the LIBERO-Spatial, LIBERO-Object, LIBERO-Goal, and LIBERO-10 suites to train both our world model and the SmolVLA policy. This dataset, comprising 1,693 trajectories with 75 to 505 frames each, is publicly available on Hugging Face (available here[2]). We conducted our evaluation exclusively on the LIBERO-10 benchmark, as its challenging long-horizon tasks provide an ideal testbed for validating the efficacy of our planning framework.

**Meta-World.** The Meta-World benchmark (Yu et al., 2021) is a comprehensive robotic manipulation suite designed for multi-task and meta-learning. We utilize the MT50 multi-task dataset, which consists of demonstrations for 50 diverse manipulation tasks spanning a wide range of horizons and

---

[1]https://github.com/gaoyuezhou/dino_wm
[2]https://huggingface.co/datasets/physical-intelligence/libero

difficulty levels. The dataset contains 2,500 episodes, each ranging from 14 to 500 frames, and is publicly available on Hugging Face (available here[3] ). This collection of demonstrations allows our world model and planning framework to learn dynamics across a broad variety of manipulation behaviors.

**LeRobot.** We collected real-world datasets for the task "pick the blue block and place it in the cup (PickPlace)" and "put the blue block in the drawer and close it" (Drawer) via teleoperation, using the LeRobot SO-101 leader-follower system with a fixed-view camera. Each dataset comprises 100 trajectories, each containing between 208 to 459 frames. During data collection for PickPlace, the cup's position remained fixed while the block was placed at 10 distinct initial positions, with 10 trajectories recorded for each. During data collection for Drawer, the drawer's initial pose remained fixed while the block was placed at 4 distinct initial positions, with 25 trajectories recorded for each. This dataset was then used to train our world model and the SmolVLA policy. For PickPlace task, the trained agent was tested on 5 novel block positions held out from the training set, with the final success rate calculated over 10 total trials (two attempts for each unseen position). For Drawer task, the trained agent was tested on 2 block positions included in the training set, with the final success rate calculates over 10 total trials (five attempts for each position). The SmolVLA policy was loaded from pretrained VLM weights then only the state projector and the action expert are trained when finetuning. The architectural settings follow SmolVLA with 0.45B parameter in the original work. We also leverage asynchronous inference pipeline of SmolVLA. Whenever the remaining action chunk ratio reaches certain threshold, robot client sends the observation from to the policy server and let the remote server compute the next action chunk in advance.

Table 6: Hyperparameters for SmolVLA.

| Name | Value |
|---|---|
| Batch size | 64 |
| Training steps | 50,000(Lerobot), 100,000(LIBERO-10, Meta-World) |
| Optimizer | AdamW |
| Learning rate | 0.0001 |
| Scheduler | Cosine |
| Warmup steps | 1,000 |
| Decay steps | 30,000 |
| Action chunk size | 50 |
| Async inference thres. | 0.5 |

## A.2 MAIN METHOD IMPLEMENTATION DETAILS

Our model is built on a pre-trained DINO (Caron et al., 2021) ViT backbone which is frozen during training. Specifically, we utilize DINO-ViT-S/16 on 224x224 image, which creates a 14×14 spatial tokens per image. Visual tokens from context frames are flattened as a input to the world model. Actions and proprioceptive vectors are linearly projected on a dimension of 10 respectively and concatenated to every visual token along the feature dimension axis. The world model is a causal transformer decoder, which predicts the tokens of the next timestep. The transformer has 6 attention layers, 16 attention heads per layer, and 2048 embedding dimensions. The world model is trained with Adam optimizer at a learning rate of 5e-4 without scheduler for 100 epochs, batch size of 32, dropout probability of 0.1. All experiments were conducted using at most 4 Nvidia RTX 3090s.

## A.3 PLANNING DETAILS

MPC rollouts utilize the CEM with action distribution defined as a Gaussian distribution. CEM is performed with planning horizon of 5, optimizing by mean and variance of top-10 action sequences among 100 candidate sequences. CEM optimization is done for 10 iterations and returns the mean of the action distribution as the final action sequence, executing all the planned action sequence for every MPC step. MPC iterates until the task is completed or until it reaches a maximum limit of 10 iterations.

---

[3]https://huggingface.co/datasets/lerobot/metaworld_mt50

For the PointMaze, Wall, PushT, and Block Pushing environment, the MPC horizon is set to $H = 5$. For each optimization step, we sample 100 candidate actions and select the top 10 based on the distance computed between the current and goal observations. Note that in sparse imagination scenarios, the distance is computed using only a subset of patches rather than the full patches. Each CEM optimization is performed for 10 steps. For the Wall and Block Pushing environments, the maximum number of MPC iterations is set to 10, while for Pointmaze and PushT, we set the maximum iterations to 15. For the Granular and Rope environments only employ the CEM approach for 10 steps.

## A.4 HSIC Analysis Details

**Hilbert–Schmidt Independence Criterion (HSIC).**  We measure statistical dependence between two random variables $X$ and $Y$ using the Hilbert–Schmidt Independence Criterion (HSIC) (Gretton et al., 2005), defined via kernels $k$ and $\ell$ on their respective domains. Let $K$ and $L$ be the $n \times n$ Gram matrices with entries $K_{ij} = k(x_i, x_j)$ and $L_{ij} = \ell(y_i, y_j)$, and $H = I_n - \frac{1}{n}\mathbf{1}\mathbf{1}^\top$ the centering matrix. Then the empirical HSIC is

$$\widehat{\text{HSIC}}(X, Y) = \frac{1}{(n-1)^2} \text{tr}(KHLH),$$

and we normalize it as

$$\text{nHSIC}(X, Y) = \frac{\widehat{\text{HSIC}}(X, Y)}{\sqrt{\widehat{\text{HSIC}}(X, X)\,\widehat{\text{HSIC}}(Y, Y)}},$$

so that $\text{nHSIC} \in [0, 1]$, with 0 indicating independence and 1 maximal dependence.

**Experimental Setting.**  We compute nHSIC between visual tokens and corresponding environment state vectors under varying dropout probabilities. A single dropout mask is fixed within each batch of size $n = 128$. Then, we repeat with 20 independent masks of same dropout rate and report the mean nHSIC. Visual tokens use a linear kernel, while state vectors use an RBF kernel whose bandwidth is set by the median heuristic. We utilized state vectors from the Granular dataset, consisting of the x and y coordinates for 12,774 particles.

## A.5 Attentive Probing Details

**Attentive Probing.**  Attentive probing is a nonlinear probing method priorly used for downstream tasks since visual tokens do not satisfy linear separable guarantee (Bardes et al., 2024). We use attentive probing to fairly compare predictive power of varying number of tokens. Instead of linear probing, attentive probing uses cross-attention layer to pool all input features to a single learnable query vector. After query residual connection, output is fed into 2-layer MLP with GeLU and LayerNorm, projecting to original input dimension. Then, final linear layer predicts the given probing target.

**Experimental Setting.**  Using attentive probing to predict the environment state with visual tokens under varying dropout probabilities, we present the mean and standard deviation of validation loss when training the model for 5 times with different dropout masks for each dropout rate. We also add leave-1 case where extreme dropout is performed to leave only a single random visual token. We use feature dimension of 384 and 4 heads for the cross-attention layer and hidden dimension of 4 times the feature dimension for the MLP layer. We trained the probing module for 500 epochs. The optimization was performed using Adam with a learning rate of $1 \times 10^{-5}$ and a batch size of 128, incorporating a cosine scheduler and omitting a warmup period. Parallel to our HSIC analysis, we used Granular dataset state information. Rather than attempting to precisely predict 12,774 particle locations (impractical due to the order of particles and dimensionality), our probing module predicts the mean and standard deviation of x and y coordinates for all particles, thereby estimating their approximate position and spread as state information.

## A.6 Token Selection Methods Details

**LHS.**  Latin Hypercube Sampling (LHS; (McKay et al., 2000)) is used as a simple uniform baseline that enforces spatial coverage. We treat the $H \times W$ patch grid as a 2D domain and partition it into $K$

strata corresponding to the desired number of tokens. At each step, we sample one patch from each stratum, which yields a set of retained tokens that are approximately uniformly distributed across the image.

**LTRP.** LTRP (Learning to Rank Patches; Luo et al. (2024)) utilizes a pre-trained MAE (Masked Autoencoder; He et al. (2022)) to learn a ranking model that assesses the importance of individual patch tokens. With 90% of the patch tokens masked, LTRP compares the reconstruction results (using $L_1$ distance) when each remaining token is individually removed versus when it is retained. Tokens contributing more significantly to the reconstruction error are considered more important. The ranking model is trained using ListMLE (Xia et al., 2008), optimizing the order of the top 20 items within each list. For the ranking model, we employed a ViT-Small architecture, while the MAE utilized a ViT-Base checkpoint. The ranking model was trained on ImageNet-1K for 50 epochs using four NVIDIA RTX 3090 GPUs. For aspects not explicitly mentioned, training configurations adhered to the defaults outlined in the original paper. The LTRP code is available here[4], and the MAE ViT-Base checkpoint is from here[5].

**Attention-Encoder.** Similar to Liang et al. (2021); Wang et al. (2022); Zhang et al. (2024b), we gauge token importance by examining the attention map between the CLS token and other patch tokens within a pre-trained DINO encoder. Tokens receiving higher attention weights from the CLS token are deemed most significant. We implement token dropping by retaining the top-K tokens sorted by their attention weights. Our experiments utilized the DINO-ViT-S/16 checkpoint[6], specifically leveraging the attention map from the final transformer block of the DINO encoder.

**STAR.** STAR (SynergisTic pAtch pRuning; Zhang et al. (2024b)) determines a comprehensive importance score for each layer by combining two distinct components: an intra-layer importance score, derived from the attention weights between the CLS token and individual patch tokens, and an inter-layer importance score, quantified using Layer-wise Relevance Propagation (LRP). The inter-layer scores were pre-computed on the ImageNet-1K dataset, consistent with the methodology outlined in the original publication. Given that the original STAR implementation was based on DeiT, we modified their codebase to ensure compatibility with DINO. The aggregated score for layer $l$, denoted $S_l$, is formulated as $S_l = S_l^{\text{intra}^\alpha} \times S_l^{\text{inter}^{(1-\alpha)}}$. We set $\alpha = 0.3$, which is one of the values proposed in the original paper. Analogous to the Attention-Encoder strategy, token dropping was realized by sorting tokens according to their scores from the final encoder layer in descending order and preserving only the top-K. The official code of STAR can be found here [7].

**Attention-WM.** Unlike encoder-based methods like Attention-Encoder, this approach stems from the idea that it might be more efficient to retain tokens deemed important by the world model during dynamics prediction for a given action. Thus, we leverage the attention maps from our trained ViT-based world model. Since our world model does not utilize a CLS token, we instead consider the attention maps involving all individual patch tokens. Specifically, for each patch token acting as a key, we calculate the average attention it receives from all query patch tokens. We then assume that tokens with higher average attention are more important. A drawback of this method is the two-pass forward computation during planning. First, the world model must be forwarded with all tokens to obtain the attention maps across all layers. These maps are then used as importance scores to select the most significant patch tokens. Subsequently, a second forward pass is executed with only the selected tokens, introducing an additional computational cost. In this approach, we averaged attention maps across all layers.

**Agglomerative Token Clustering (ATC).** Inspired by Haurum et al. (2024), we employ a modified version of Agglomerative Token Clustering (ATC) for our clustering approach. Necessary modifications were made as the original ATC was primarily designed for single-frame applications, such as classification or segmentation. Adapting it for goal-conditioned world model-based planning necessitated a modified mechanism to address multiple history frames, and to match clusters between the goal image features and predicted latent features.

---

[4] https://github.com/irsLu/ltrp
[5] https://github.com/facebookresearch/mae
[6] https://github.com/facebookresearch/dino
[7] https://github.com/LannWei/STAR_ICLR

Our clustering methodology proceeds as follows. First, we extract DINO features from the input image and apply agglomerative clustering using either $L_2$ distance or cosine similarity (the choice between these two metrics did not significantly affect results). The number of clusters is dynamically determined by the desired drop ratio (e.g., 137 clusters for a 20% drop). Using the cluster assignments from the initial frame as anchors, subsequent frames undergo K-means clustering. After clustering, patches within each cluster are aggregated via average pooling, maintaining the same number of patches as clusters. To address inconsistencies in clustering results across visually similar scenes, we observed through trial and error that storing and re-using K-means centroids from the previous frame as initial centroids for the subsequent frame ensures better consistency by aligning clusters across consecutive frames.

To integrate this into MPC, we initially cluster the goal image and use it as an anchor for clustering the current observation, aiming for inherent cluster alignment. However, this alone did not guarantee perfect alignment, especially with an increased number of clusters. To mitigate this, we introduced an additional matching step: during the loss calculation within the CEM, specifically when selecting the top-K actions, we employ Hungarian matching between the world model's predictions and the goal clusters to ensure proper alignment. Without this crucial matching step, we observed that the results degrade significantly. For instance, in the Wall Environment, using 128 clusters, experiments over 30 episodes showed that the success rate improved from 70.0% without matching to 86.7% with matching applied.

### A.7 Cluster Retention Entropy Details

To quantitatively measure the spatial bias of different token sampling methods, we introduce **Cluster Retention Entropy**. This metric assesses how uniformly a sampling strategy distributes its kept tokens across the spatial layout of an image. The calculation proceeds as follows:

1. **Spatial Clustering:** First, we establish a set of spatial anchors. We take all visual patch tokens from a single, random image (e.g., Block Pushing) and cluster them into $K$ groups based on their spatial coordinates using the K-Means algorithm. This gives us $K$ spatial regions of interest.

2. **Distribution of Retained Patches:** For a single image and a given sampling method, we identify the subset of patches that are *retained*. We then count how many of these retained patches fall into each of the $K$ pre-defined spatial clusters. This count creates a discrete probability distribution over the clusters.

3. **Entropy Calculation:** Finally, we compute the Shannon entropy of this distribution. A higher entropy value indicates that the retained patches are spread more uniformly across the spatial clusters, signifying low spatial bias (i.e., good coverage). Conversely, a low entropy value indicates that the sampling method is heavily biased, concentrating its attention on only a few spatial regions.

In the analysis presented in the main text, we computed the mean entropy for each method on the Block Pushing dataset. To ensure a robust comparison, we averaged the results across a range of settings, varying the number of clusters $K \in \{2, 4, 8, 12\}$ and the token drop ratio from 10% to 90%.

### A.8 Real-World LeRobot Planning Details

To perform goal-conditioned planning in the real-world LeRobot environment, we used the goal image shown in Figure 19. The corresponding goal proprioceptive state, which represents the robot's joint information, was set to $[23.19, -17.32, 12.97, 79.24, -4.68, 16.58]$ (PickPlace) and $[-31.44, 0.68, 16.94, 67.90, -26.77, 0.61]$ (Drawer).

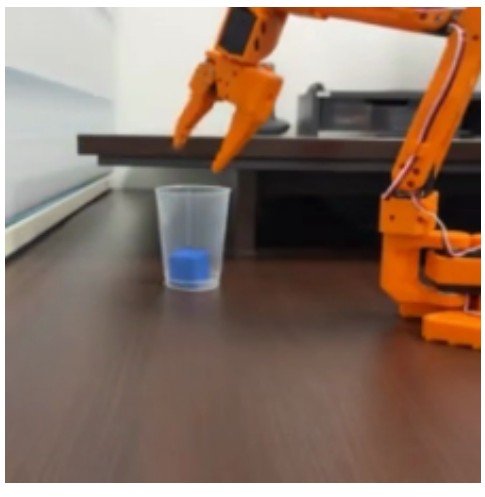 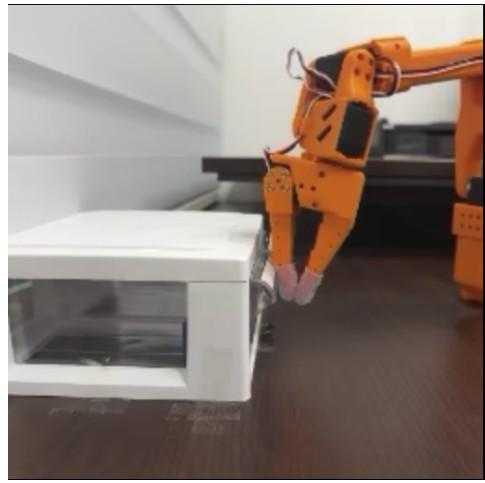

(a) PickPlace                              (b) Drawer

Figure 10: **The goal images used for planning in the real-world LeRobot environment.**

## B   ADDITIONAL ABLATIONS AND EXPERIMENTS

### B.1   HYPERPARAMETER

**CEM Optimization Steps.**   We selected 10 CEM optimization steps for consistency with the DINO-WM baseline, ensuring fair comparison. Our ablation study in the PushT environment (60 episodes, various drop ratios) validated this choice: 5 steps yielded 49.3% success, while 10 steps achieved 51.7%. Increasing to 15 steps (50.7%) offered no significant benefit. Thus, 10 optimization steps represent a well-justified trade-off between planning quality and computational cost.

**Longer Planning Horizon in MPC.**   To further assess the robustness of our approach, we evaluated its performance under a more demanding, longer planning horizon. Specifically, we conducted experiments in the PushT environment with the planning horizon extended to H=7. Under this challenging condition, the Full-Patch baseline, which utilizes all visual tokens for planning, achieved a success rate of 48.3%. Our sparse imagination method remained highly competitive, achieving a 50.0% success rate with a 20% token drop and 46.7% with a 40% drop. This demonstrates that our framework's performance does not degrade relative to the **Full-Patch** baseline even on longer-horizon tasks, confirming the robustness of planning with a sparse subset of tokens.

**Frame Skips in World Model Training.**   We performed an ablation on the frame skip value in the Pointmaze environment. We found that a frame skip of 1, which processes every frame, resulted in a low success rate of 36.7%, likely due to redundant observations. In contrast, increasing the frame skip to 5 significantly boosted performance to 85.8%. However, further increasing the frame skip to 10 yielded no additional benefit, with the success rate remaining at 85.5%. These results demonstrate a clear point of diminishing returns and justify our final choice of using a frame skip of 5, which provides an optimal balance between task performance and computational efficiency.

### B.2   GROUPED ATTENTION

This section investigates the influence of randomized grouped attention (applied during training) on world model prediction error during planning, relative to full attention (Fig. 11). The analysis spans multiple datasets and various dropout ratios. In contrast to the main text's primary focus on findings for the Wall dataset under a 50% dropout condition, the present discussion offers a more comprehensive overview of results across these diverse conditions. We quantify this impact by presenting the mean relative $L_2$ distance of retained tokens on the validation set of each dataset, both with and without dropout, for each attention mechanism. This metric normalizes the $L_2$ distance by the tokens' norm. All reported values represent the mean and standard deviation derived from

five independent experiments. Our analyses consistently demonstrate that grouped attention leads to lower prediction error and reduced variance compared to full attention across the Wall, Granular, Rope, and Block Pushing datasets. We observe an apparent correspondence between these improvements and notable enhancements in success rates across the evaluated datasets (as detailed in the main text). This observed pattern seems to hold for a majority of the dropout ratios investigated.

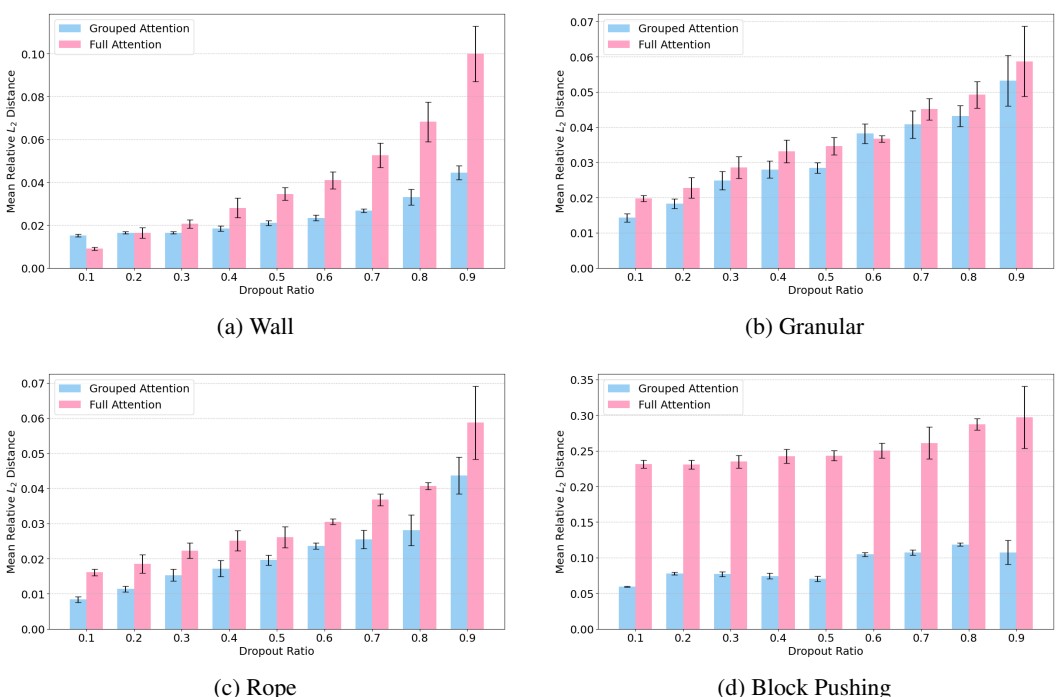

(a) Wall

(b) Granular

(c) Rope

(d) Block Pushing

Figure 11: **Contribution of grouped attention during pretraining to planning on various drop ratios across various datasets.**

## B.3 Planning Robustness to Noises

In this section, we experimentally validate our claim that the remaining prediction error, consistent with prior studies on CEM's robustness to noise and uncertainties, is sufficient for effective planning. To investigate this, we first examined whether the prediction error, incurred by token dropping in the world model, correlates with the world model's input across different dropout ratios. We reduced the high-dimensional input features to a 2D representation using UMAP and generated a scatter plot comparing these UMAP coordinates with the magnitude of the error (Frobenius norm) (Fig. 12).

Visual inspection of the UMAP projection of input features shows no obvious global or local patterns linking UMAP coordinates of model inputs to the error norm. Data points, colored by their error magnitude (Frobenius norm), are generally intermixed throughout the low-dimensional embedding. Therefore, this specific UMAP representation of input features, based on visual analysis, does not clearly indicate a strong, direct relationship between the spatial embedding of these features and their corresponding error magnitude.

Given the findings, we assume that the prediction error inherent in world model when operating on token subsets could be approximated as zero-mean Gaussian noise, exhibiting no direct correlation with the input tokens. To empirically test the robustness of MPC-CEM planning, we proceeded to add this synthesized noise to the input tokens during the actual planning.

To simulate the behavior of a dropout-affected world model, we begin by generating predictions using the full world model without applying dropout. Next, we modify these predictions by randomly dropping a subset of the predicted tokens and adding zero-mean Gaussian noise to the remaining ones. These modified predictions are then used for MPC-CEM planning. The magnitude of the Gaussian noise was scaled relative to the token norm and experimented under varying noise scales

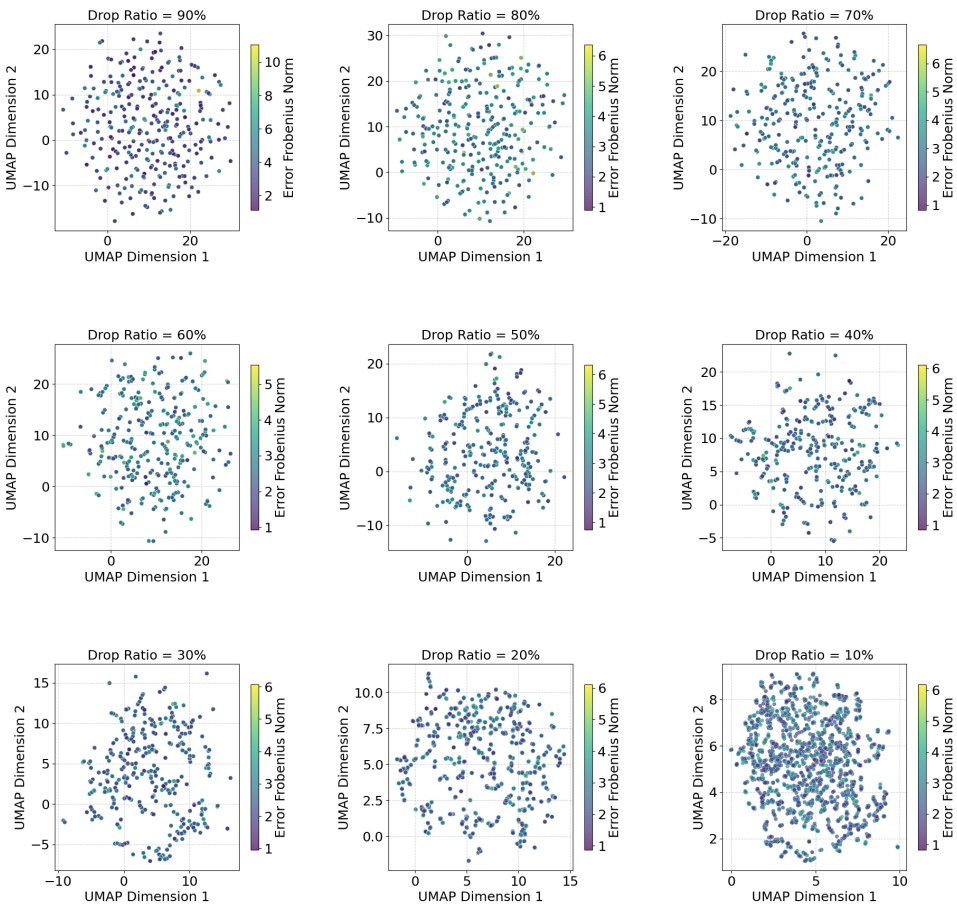

Figure 12: **Scatter plots visualizing the embedding of world model inputs in a 2D UMAP manifold, colored by their prediction error magnitude (Frobenius norm), presented across different token drop ratios.**

and dropout rates. We then evaluated the robustness of MPC-CEM planning by measuring the success rates of actual rollouts in the environment using these noisy predictions. This experiment was conducted in the point maze environment, with the agent allowed a maximum of 3 MPC planning steps per episode. The presented success rates were averaged over 60 independent evaluations.

The results (Fig. 13) indicate that MPC-CEM planning sustains effective performance at dropout levels with minimal performance decrease, which aligns with our main experiments. While its performance remains stable as noise levels increase up to a specific threshold, beyond which it begins to decline. This suggests that MPC-CEM is robust to prediction errors caused by token dropout, reinforcing our claim that manageable for planning. These observations also align with prior works on the resilience of CEM to noise and uncertainties.

## B.4   DROPOUT VS. CLUSTERING IN EXECUTION TIMES

Table 7 compares the execution times of single MPC iterations between dropout and clustering methods at various compression rates (drop ratios). The clustering approach requires additional computations, including the calculation of pairwise distances or similarities among patches, incurring a quadratic complexity relative to the number of clusters. Furthermore, since clustering merges patches from multiple spatial locations, the resulting pooled features lose fixed spatial positions, necessitating additional matching procedures when computing distances between goal image patches and observations during MPC planning. Consequently, the clustering method requires significantly

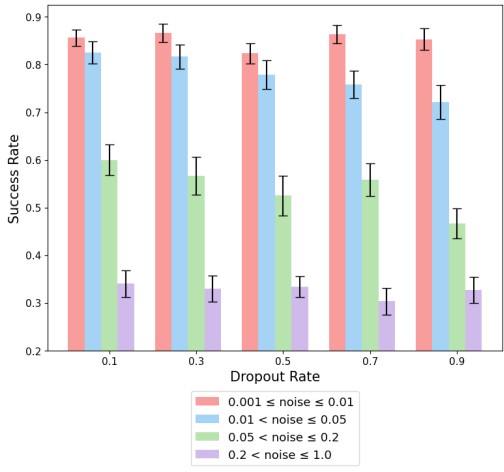 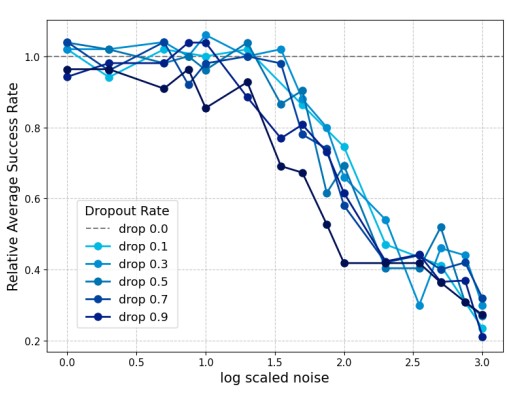

(a) MPC-CEM success rate vs. dropout rate under various noise conditions.

(b) Relative MPC-CEM success rate vs. log noise scale for different dropout ratios.

Figure 13: **Robustness of MPC-CEM planning to prediction dropout and noise. The results show minimal degradation of success rates by increasing dropout rates and performance loss only after a certain noise threshold.**

Table 7: Comparison of planning time and change across different drop ratios (compression rates) between drop and clustering apporaches.

| Drop Ratio (Compression Rate) | Pointmaze Planning Time (s/iter) Drop | Pointmaze Planning Time (s/iter) Cluster | Change (%) | Wall Planning Time (s/iter) Drop | Wall Planning Time (s/iter) Cluster | Change (%) | PushT Planning Time (s/iter) Drop | PushT Planning Time (s/iter) Cluster | Change (%) | Block Pushing Planning Time (s/iter) Drop | Block Pushing Planning Time (s/iter) Cluster | Change (%) |
|---|---|---|---|---|---|---|---|---|---|---|---|---|
| 10% | 165 | 332 | +101.2 | 73 | 246 | +237.0 | 149 | 328 | +120.1 | 278 | 478 | +71.9 |
| 20% | 141 | 297 | +110.6 | 69 | 235 | +240.6 | 131 | 295 | +125.2 | 259 | 444 | +71.4 |
| 30% | 126 | 269 | +113.5 | 65 | 235 | +261.5 | 114 | 267 | +134.2 | 240 | 407 | +69.6 |
| 40% | 106 | 245 | +131.1 | 62 | 207 | +233.9 | 97 | 237 | +144.3 | 214 | 376 | +75.7 |
| 50% | 93 | 214 | +130.1 | 53 | 192 | +262.3 | 82 | 213 | +159.8 | 208 | 359 | +72.6 |
| 60% | 80 | 193 | +141.3 | 50 | 184 | +268.0 | 69 | 187 | +171.0 | 200 | 318 | +59.0 |
| 70% | 69 | 172 | +149.3 | 46 | 164 | +256.5 | 59 | 170 | +188.1 | 184 | 295 | +60.3 |
| 80% | 56 | 152 | +171.4 | 46 | 145 | +215.2 | 49 | 139 | +183.7 | 175 | 266 | +52.0 |
| 90% | 48 | 137 | +185.4 | 42 | 117 | +178.6 | 38 | 115 | +202.6 | 167 | 248 | +48.5 |

more computation, with runtime increases of up to 268.0% compared to dropout, as shown in Table 7. In contrast, our dropout approach merely samples patch indices, which results in nearly constant computational overhead regardless of dropout ratio. Additionally, since the same patch subsets are consistently applied to both observation and goal images, the spatial positions remain fixed, eliminating the need for extra matching processes.

## B.5 GENERALIZATION TO DIFFERENT PLANNING PARADIGMS

To demonstrate that our sparse imagination framework is not limited to sampling-based planning algorithms like the CEM, we evaluate its compatibility with a gradient-based trajectory optimizer (MPC-GD). We replace the CEM planner with a gradient descent-based optimizer in the PointMaze and Wall environments, while keeping the randomized grouped attention and sparse token selection mechanisms identical to the main experiments.

Table 8 presents the planning success rates across varying dropout ratios. The results indicate that our method maintains high performance even when utilized with gradient-based optimization. Notably, with a 50% token drop, the success rates remain comparable to the Full-Patch baseline (e.g., 94.0% vs. 94.0% in PointMaze, and 88.0% vs. 88.0% in Wall). This suggests that random token sparsification preserves the optimization landscape sufficiently for gradient-based planners to con-

verge to effective solutions, confirming the general applicability of our approach across different planning paradigms.

Table 8: Performance of Sparse Imagination using a Gradient-Descent based Planner (MPC-GD) on PointMaze and Wall environments (Mean Success Rate, %).

| Drop Ratio | PointMaze | Wall |
|---|---|---|
| Full (0%) | 94.0 | 88.0 |
| 10% | 94.0 | 90.0 |
| 30% | 98.0 | 92.0 |
| 50% | 94.0 | 88.0 |
| 70% | 90.0 | 82.0 |
| 90% | 78.0 | 78.0 |

### B.6 EXPLORING UNCERTAINTY-AWARE ADAPTIVE DROPOUT

While our main experiments characterize performance as a function of fixed sparsity levels (demonstrating robustness up to approximately 50% dropout), we do not claim that a single fixed ratio is universally optimal. In scenarios where the optimal drop ratio is unknown or task complexity fluctuates, an online adaptive mechanism becomes desirable. Since our world model is explicitly trained to be robust under random sparsification via randomized grouped attention, the Sparse Imagination framework naturally supports dynamic adjustment of the dropout ratio at test time without retraining.

To explore this capability, we implement a simple uncertainty-aware adaptive planner. We utilize the variance of latent rollouts across MPC samples as a proxy for prediction uncertainty. The logic is straightforward: if uncertainty increases between iterations, the planner reduces the drop ratio (retaining more tokens); if uncertainty is low, the drop ratio is increased to save computation.

We evaluate two variants: (1) **Ada-Rand**, which updates the scalar drop ratio based on the change in variance and resamples a new random mask; and (2) **Ada-Inc**, which updates the mask incrementally, preferentially keeping high-variance tokens. Table 9 summarizes the results. In the Wall environment, Ada-Rand achieves a 95.0% success rate (matching the strong fixed 50% baseline) with an average drop ratio of 26.1%. In PushT, Ada-Rand reaches 68.3% success (matching the 10–50% fixed range). Crucially, these adaptive methods avoid the sharp performance collapses observed with overly aggressive fixed sparsity (e.g., >70% in PushT) by automatically settling at a safe sparsity level. While they do not necessarily outperform the best-tuned fixed ratio, they demonstrate the potential of our framework to support online optimization when the optimal sparsity is unknown.

Table 9: Feasibility study of Uncertainty-Aware Adaptive Planning variants (Mean Success Rate, %) and the resulting Average Drop Ratio (%).

| Method | Wall | | PushT | |
|---|---|---|---|---|
| | Success Rate | Avg. Drop | Success Rate | Avg. Drop |
| Ada-Rand | 95.0 | 26.1 | 68.3 | 29.0 |
| Ada-Inc | 91.7 | 29.8 | 65.0 | 35.8 |

### B.7 WORLD MODEL PREDICTION QUALITY

To assess whether our training mechanism affects the intrinsic quality of the world model's predictions, we train the world models with an optional decoder and measure the image reconstruction quality of the predicted features using the Learned Perceptual Image Patch Similarity (LPIPS) metric (Zhang et al., 2018) (lower is better). Table 10 compares the LPIPS scores of the standard Full (full attention) baseline against our model trained with Randomized Grouped Attention ("Drop"), where both models decode from full patches at evaluation time.

Across various environments, the LPIPS values for our method (Drop) remain very close to the Full-Patch baseline. Notably, in visually or physically complex tasks such as Granular and Rope, our

method achieves slightly better reconstruction scores, potentially due to the regularization effect of the grouped attention strategy. These results indicate that our training mechanism does not degrade the world model's prediction quality under full-patch evaluation while enabling robust performance under sparse imagination at planning time.

Table 10: Comparison of World Model Prediction Quality (LPIPS, lower is better).

| Environment | Full | Drop (Ours) |
|---|---|---|
| PointMaze | 0.00028 | 0.00031 |
| Wall | 0.00074 | 0.00078 |
| Granular | 0.05607 | 0.04739 |
| Rope | 0.03347 | 0.01906 |
| PushT | 0.00561 | 0.00644 |

## B.8 DETAILED PLANNING TIME ANALYSIS

To investigate the source of computational efficiency, we profile the latency of major components within the planning loop. Table 11 presents the fine-grained latency breakdown for the PushT environment.

The analysis confirms that the dominant acceleration stems from the World Model's **Self-Attention** mechanism. As shown in the tables, Self-Attention latency drops dramatically (e.g., in PushT: $100\% \rightarrow 28.8\%$ at 50% drop), closely following the expected quadratic reduction. In contrast, the Projection and MLP layers scale linearly, while fixed costs remain stable.

*Note: The latency values reported here were measured in a separate additional profiling session and may exhibit minor variances compared to the end-to-end timings in Table 2.*

Table 11: Latency breakdown per planning iteration on **PushT**. Format: **Time in seconds (Reduction % compared to Full baseline)**.

| Drop (%) | Variable Costs (World Model) | | | Fixed Costs | | |
|---|---|---|---|---|---|---|
| | Self-Attn | Proj (QKV) | MLP (FFN) | Encoder | Env Step | Others |
| 0 | 87.4 (-0.0%) | 30.5 (-0.0%) | 34.5 (-0.0%) | 11.9 | 2.8 | 10.8 |
| 10 | 73.3 (-16.0%) | 27.5 (-9.9%) | 31.1 (-9.7%) | 11.7 | 3.4 | 10.3 |
| 20 | 57.2 (-34.6%) | 24.6 (-19.2%) | 27.9 (-19.0%) | 11.9 | 4.7 | 11.1 |
| 30 | 47.0 (-46.2%) | 21.6 (-29.2%) | 24.5 (-28.8%) | 11.9 | 3.4 | 10.6 |
| 40 | 32.9 (-62.3%) | 18.7 (-38.7%) | 21.1 (-38.7%) | 11.8 | 3.5 | 11.6 |
| 50 | 25.1 (-71.2%) | 15.5 (-49.3%) | 17.5 (-49.2%) | 11.8 | 2.8 | 10.3 |
| 60 | 16.0 (-81.7%) | 12.5 (-59.2%) | 14.0 (-59.3%) | 11.8 | 4.0 | 10.5 |
| 70 | 10.6 (-87.9%) | 9.5 (-68.8%) | 10.7 (-69.1%) | 11.8 | 2.4 | 11.1 |
| 80 | 5.0 (-94.3%) | 6.4 (-79.2%) | 7.1 (-79.3%) | 11.8 | 3.3 | 10.4 |
| 90 | 2.4 (-97.2%) | 3.5 (-88.5%) | 3.8 (-88.9%) | 11.7 | 3.4 | 10.3 |

## B.9 ADDITIONAL ANALYSIS ON BLIND SPOTS

To provide a quantitative characterization of the "blind spot" phenomenon visualized in Figures 9 and 20, we perform a controlled analysis on the PushT environment. At each MPC planning iteration, we track the ground-truth position of the agent (blue ball) for all candidate action sequences. We project the agent's position into the image space. We approximate the blue ball as a circle whose diameter equals one patch size and use this circle to determine which patch tokens it overlaps. A blind-spot event is counted if *all* tokens overlapping with the agent are dropped by the sampling mask for a given timestep.

Table 12 presents the blind-spot occurrence rates evaluated at 50% and 60% dropout ratios (60 episodes). At 50% drop, Random sampling produces blind spots in only **3.10%** of states, whereas baseline methods fail to capture the agent in 17–26% of cases. At 60% dropout, while failure rates increase due to higher sparsity, Random sampling (**12.26%**) still maintains a lower blind-spot rate

compared to baselines (22–33%). These results quantitatively demonstrate that Random sampling consistently yields fewer blind-spot events compared to other selection strategies.

Table 12: Quantitative analysis of Blind Spot occurrence rates (%) in the PushT environment (60 episodes). Lower is better.

| Method | 50% Drop | 60% Drop |
|---|---|---|
| **Random (Ours)** | **3.10** | **12.26** |
| STAR | 17.74 | 22.87 |
| LTRP | 18.56 | 27.05 |
| Attention-Encoder | 20.99 | 33.04 |
| Attention-WM | 26.60 | 31.17 |

# C VISUAL DEMONSTATIONS

## C.1 IMAGINED ROLLOUT BY WORLD MODEL

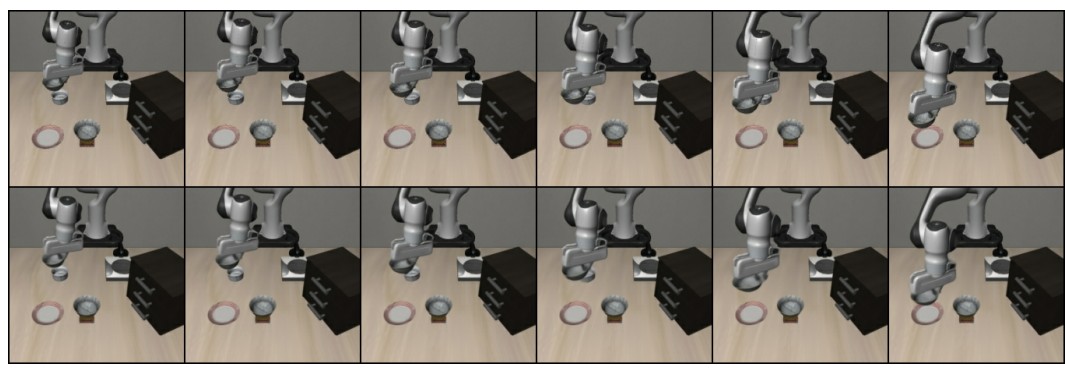

(a) LIBERO

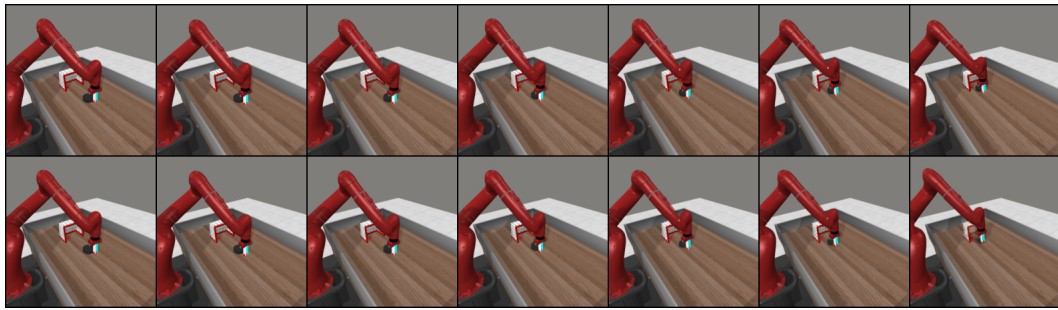

(b) Meta-World

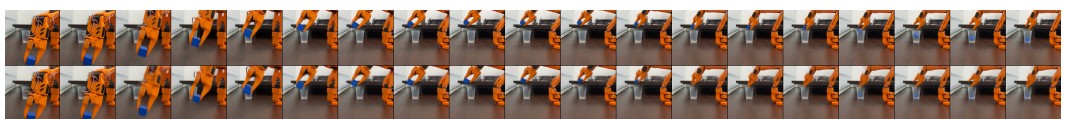

(c) LeRobot PickPlace

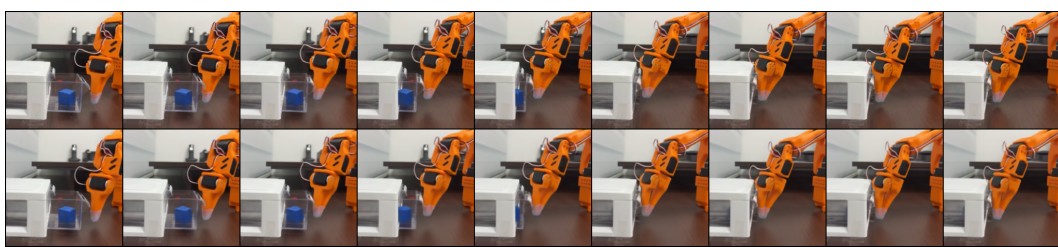

(d) LeRobot Drawer

Figure 14: **Imagined rollout of the LIBERO, Meta-World, real-world LeRobot Pickplace and Drawer by world model. Top row: Ground truth. Bottom row: Imagined rollout by world model.**

### C.2 EXECUTED ROLLOUT DURING PLANNING

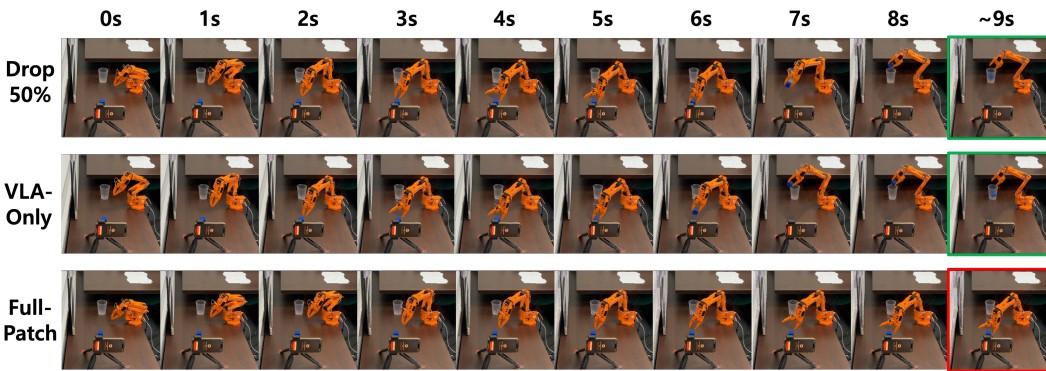

Figure 15: **Executed rollout demonstration on real-world PickPlace task in LeRobot environment (50% Drop, VLA-only, and Full-Patch).** A green border indicates a successful episode, while a red border indicates incompleteness. Both our method and the VLA-only baseline succeed in under 9 seconds, while the Full-Patch planner fails to do so in the same timeframe due to its significant planning latency.

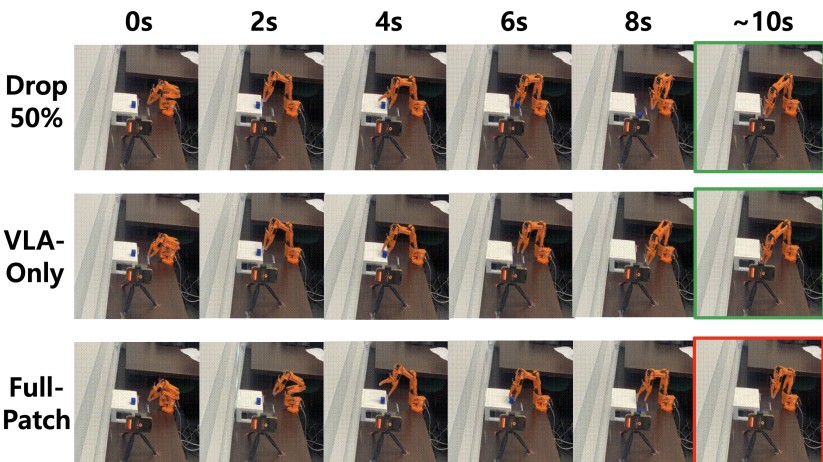

Figure 16: **Executed rollout demonstration on real-world Drawer task in LeRobot environment (50% Drop, VLA-only, and Full-Patch).** A green border indicates a successful episode, while a red border indicates incompleteness. Both our method and the VLA-only baseline succeed in under 10 seconds, while the Full-Patch planner fails to do so in the same timeframe due to its significant planning latency.

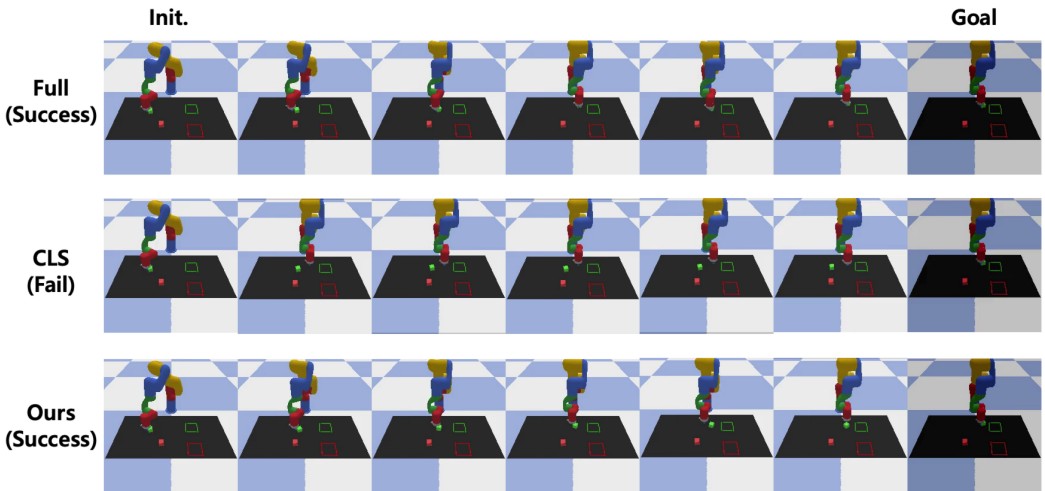

Figure 17: **Executed rollout demonstration on Block Pushing environment of Full, CLS, and our methods (60% Drop).**

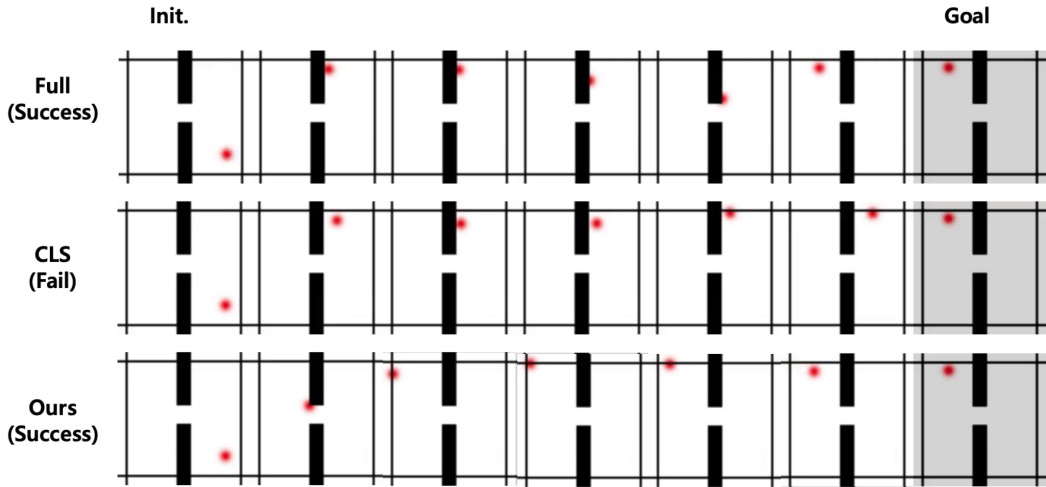

Figure 18: **Executed rollout demonstration on Wall environment of Full, CLS, and our methods (50% Drop).**

## C.3    ADDITIONAL BLIND SPOT EXAMPLES

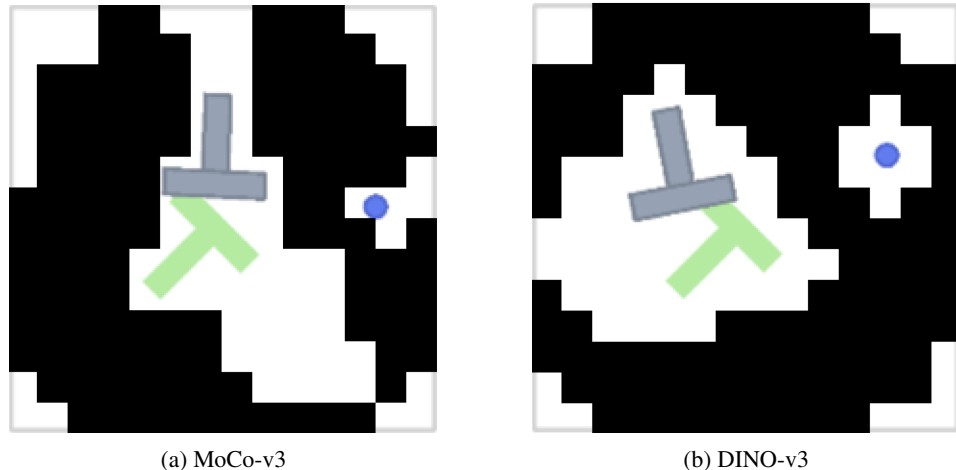

(a) MoCo-v3                                          (b) DINO-v3

Figure 19: **Visualization Examples of the Blind Spot Problem in Additional Visual Encoders (60% Drop in PushT).**

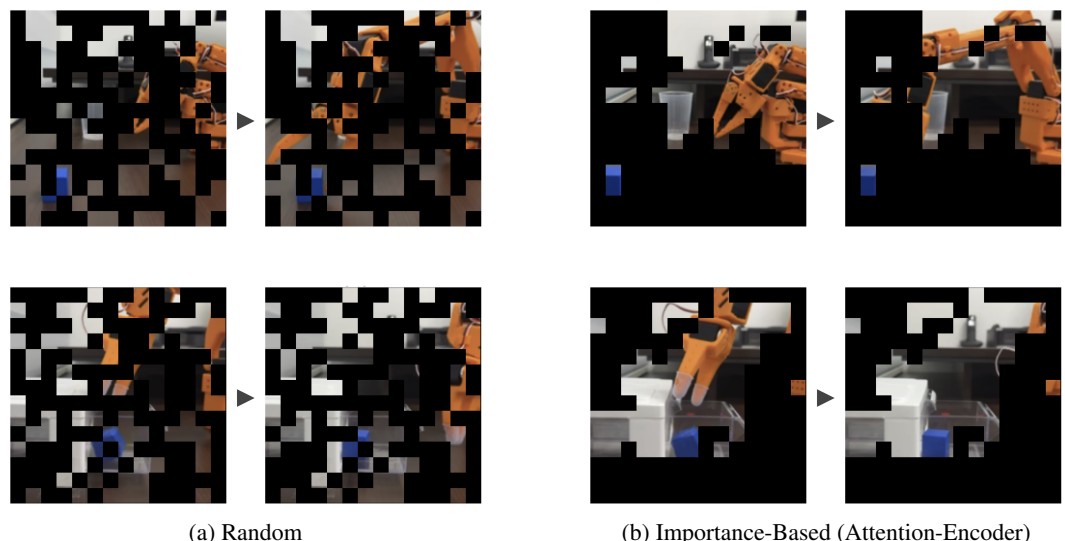

(a) Random                              (b) Importance-Based (Attention-Encoder)

Figure 20: **Visualization Example of the Blind Spot Problem in Real-World LeRobot Environment.**

## D    LLM USAGE

LLMs were utilized solely for improving the clarity, grammar, and style of the paper and supplementary materials. No LLM was used for research ideation, experimental design, data generation, or result analysis. The authors take full responsibility for the content of this submission.

