# OpenReview forum: "Sparse Imagination for Efficient Visual World Model Planning"
_ICLR.cc/2026/Conference — ICLR 2026 Poster_

### Official Review · Reviewer_gLWK · 2025-10-27

**Soundness:** 3
**Presentation:** 3
**Contribution:** 3
**Rating:** 6
**Confidence:** 2

**Summary:**

This paper proposes Sparse Imagination, a simple yet effective approach for improving computational efficiency in visual world-model-based planning.The key idea is to perform model predictive control (MPC) rollouts on only a random subset of ViT patch tokens, rather than processing all of them.To make this feasible, the authors train a transformer world model with a randomized grouped-attention strategy, enabling robustness to missing visual tokens at inference time.
During planning, a random dropout mask is applied dynamically at each iteration, which substantially reduces the quadratic computational cost of self-attention.Extensive experiments on seven simulated environments (e.g., Pointmaze, PushT, Rope, LIBERO-10) and a real-world LeRobot setup show that the method achieves comparable task success to the full-token baseline while reducing planning time by more than 50%.

**Strengths:**

1. **Clear motivation and simplicity**.
The paper tackles an important bottleneck of visual world-model planning (quadratic attention cost) with a conceptually simple and practical solution.

2. **Strong empirical validation**.
Experiments are extensive, covering both simulation and real robotic control tasks. Results convincingly demonstrate significant efficiency gains with minimal performance loss.

**Weaknesses:**

1. **Limited algorithmic novelty**.
While effective, the proposed approach mainly combines known components (token dropout + grouped attention) and lacks deeper theoretical or architectural innovation.

2. **Shallow theoretical grounding**.
The work frames the idea as “redundancy reduction,” yet offers little formal analysis or justification of when and why sparse imagination preserves sufficient task information.

3. **Empirically chosen design choices**.
The dropout ratio and grouping strategy are empirically chosen; no adaptive mechanism or general rule for balancing performance and speed is proposed.

**Questions:**

1. Can the dropout ratio be made adaptive to uncertainty or task complexity?
2. Does grouped attention reduce long-range spatial reasoning ability?
3.  What is the real-world end-to-end latency improvement on hardware?

---

> ### Author Response · Authors · 2025-11-21
>
> We thank the reviewer for the constructive feedback and insightful questions.
>
> **W1. Limited algorithmic novelty. While effective, the proposed approach mainly combines known components (token dropout + grouped attention) and lacks deeper theoretical or architectural innovation.**
>
> We respectfully clarify that our core contribution goes beyond a straightforward combination of existing techniques, and we see the novelty at both the **system level** and the **counter-intuitive insight** level.
>
> - **System-level mechanism.** At training time, we use Randomized Grouped Attention to make the world model intrinsically robust to arbitrary missing tokens. At test time, Sparse Imagination turns this robustness into a simple *training-free sparsity knob*. Empirically, in our benchmarks, sparsity can be increased up to around 50% with only mild impact on success rates (Table 1, Fig. 5), while yielding substantial reductions in planning latency.
>
> - **Counter-intuitive insight.** To the best of our knowledge, a central and novel finding is that **simple random sampling consistently matches or outperforms complex importance-based schemes** in planning. Our analysis in Sec. 5.4 shows that static importance maps create persistent blind spots in downstream planning, while Random sampling avoids these failures with negligible overhead. This runs counter to the common intuition that importance-based methods are always preferable, and we believe this empirical insight itself is a key contribution.
>
> **W2. Shallow theoretical grounding. The work frames the idea as “redundancy reduction,” yet offers little formal analysis or justification of when and why sparse imagination preserves sufficient task information.**
>
> We agree that our current work stops short of a full formal theory, but we did make a deliberate effort to clarify *why* sparse imagination works and *when* it starts to fail through several diagnostic analyses. In particular, (i) our nHSIC and attentive probing studies (Sec 5.3) quantify how much task-relevant state information is redundantly distributed across patch tokens under different dropout rates; (ii) our blind-spot and coverage analyses (Sec 5.4) compare static importance-based masks against unbiased Random sampling, highlighting how static masks can permanently remove parts of the workspace and collapse feature variance. Together, these analyses provide an intuitive picture of the mechanism behind sparse imagination.
>
> Also, we provide an additional intuitive explanation of the observed success patterns using visual redundancy and task difficulty.
>
> **Visual redundancy.**
>
> We compare Wall and PointMaze, which are both 2D navigation tasks with similar control structure but markedly different visual textures: Wall has a simple, mostly uniform background, whereas PointMaze uses a high-frequency checkerboard pattern. At a 90% drop ratio, Wall still achieves 86.7% success, while PointMaze drops to 71.7%. One plausible interpretation is that in visually simpler scenes, similar information is more redundantly distributed across patches, so planning can tolerate more aggressive sparsification, whereas in highly textured scenes the same drop ratio more readily degrades the performance.
>
> **Task difficulty.**
>
> Wall and PushT share relatively clean, top-down visuals (similar scene complexity), but Wall is a relatively simple navigation task whereas PushT involves more challenging, fine-grained manipulation. Wall tolerates very high sparsity and retains near 90% success even at large drop ratios, whereas PushT begins to degrade more noticeably beyond 50% dropout. This suggests that more difficult tasks that depend on subtle geometric details require a higher effective token density.
>
> These explanations do not imply that visual redundancy and task difficulty are the only factors at play; other aspects like scene observability may also affect sensitivity to sparsification. **Our goal here is not to provide a complete theory, but to add intuitive structure on top of the experimental results, clarifying when sparse imagination worked well in our benchmarks and where it started to break down.**

---

> ### Author Response · Authors · 2025-11-21
>
> **W3. Empirically chosen design choices. The dropout ratio and grouping strategy are empirically chosen; no adaptive mechanism or general rule for balancing performance and speed is proposed.**
>
> We appreciate the opportunity to clarify the motivations behind these choices and to discuss adaptation.
>
> **Grouped Attention as a structural design, not a tuned hyperparameter.**
>
>    Grouped Attention is introduced to *structurally* enforce robustness to missing tokens: by randomly partitioning tokens into groups and restricting attention within each group at training time, the model is explicitly trained to operate under partial visual contexts. This is not tuned per-environment; rather, it is a design choice that enables test-time sparsification without retraining.
>
> **Dropout ratio and performance.**
>
>    We do not present or tune a single “optimal” drop ratio. Instead, we sweep over a range of sparsity levels and report how success rate and planning time vary as a function of the drop ratio across environments (Tables 1–2). In our benchmarks, performance remains close to the full-patch setting up to around 50% drop while latency decreases substantially. Our goal here is to characterize this empirical trade-off curve, rather than to introduce an optimal drop ratio.
>
> **Uncertainty-aware adaptive sparsity (new experiment).**
>
>    To further address the reviewer’s concern, we implemented a simple uncertainty-aware adaptive planner. The core idea is to adjust the token drop ratio based on the variance of latent rollouts across MPC samples, using this variance as a proxy for prediction uncertainty: when uncertainty increases between iterations, the planner automatically reduces the drop ratio (i.e., keeps more tokens), and when uncertainty is low, it can safely increase drop ratio.
>
> For the adaptive mechanism, we evaluated two uncertainty-aware variants, Ada-Rand and Ada-Inc. In Ada-Rand, at each MPC iteration we (i) compute the variance of latent rollouts across elite trajectories, (ii) compare it to the previous iteration, and (iii) update the drop ratio with this change in variance (higher uncertainty → lower drop ratio, lower uncertainty → higher drop ratio). We then sample a new mask according to the updated drop ratio
>
> In Ada-Inc, we use the same variance signal but update the mask incrementally rather than resampling from scratch. Starting from the previous mask, tokens with persistently low variance across elites are gradually dropped, while tokens whose latent features exhibit high variance are preferentially added or retained. When the variance signal indicates that the keep ratio should increase (i.e., we need to reduce sparsity), we randomly revive tokens from the previously dropped set and bring them back into the mask, so that sparsity can be adjusted without permanently excluding any region.
>
> In the Wall environment, **Ada-Rand** achieved a 95.0% success rate with an average drop ratio of 26.1%, matching the performance of a strong fixed 50%–drop baseline, while **Ada-Inc** achieved 91.7% success at 29.8% average drop. In PushT, **Ada-Rand** reached 68.3% success (29.0% average drop), and **Ada-Inc** reached 65.0% (35.8% average drop). They match the performance of the best fixed drop ratios (e.g., 10% or 50% fixed drop), and they avoid the sharp performance collapses observed for overly aggressive fixed sparsity.
>
> Our goal here is not to claim that a particular fixed or adaptive schedule is universally optimal, but to show that our framework readily supports lightweight, uncertainty-based adaptation without additional training. In particular, because the world model is explicitly trained to be robust under random sparsification (via randomized grouped attention), the same Sparse Imagination mechanism can be reused at test time to adjust the drop ratio online rather than redesigning the planner or retraining the model. The fixed-ratio results characterize how performance behaves as a function of sparsity (robust up to around 50% in our benchmarks), and the adaptive variants illustrate that the same mechanism can be applied when the optimal drop ratio is unknown or varying, which we see as a promising direction for future work.
>
> **Q1. Can the dropout ratio be made adaptive to uncertainty or task complexity?**
>
> Please see our responses to W3.

---

> ### Author Response · Authors · 2025-11-21
>
> **Q2. Does grouped attention reduce long-range spatial reasoning ability?**
>
> We do not directly measure “long-range spatial reasoning” with a dedicated metric, but we can assess its effect indirectly through our downstream tasks. In practice, we do **not** observe degradation in such settings. Grouped Attention is applied with **random group assignments** at each iteration, so no pair of regions is permanently isolated: across layers and time, different tokens frequently co-occur in the same group and can still exchange information. Empirically, our method maintains high performance on environments that inherently require long-range coordination (e.g., Rope, LIBERO-10, and pushing-based tasks where global geometry and contact relationships matter), which suggests that the model retains sufficient long-range reasoning capacity under this training scheme.
>
> **Q3. What is the real-world end-to-end latency improvement on hardware?**
>
> Figure 5 and Sec. 4.3 report the real-world latency measurements on the LeRobot setup. Our method reduces the **end-to-end episode time** from 19.1 s (VLA-only baseline) to 10.4 s, corresponding to roughly a **2× speedup** in overall execution time. This includes both world-model rollouts and real robot execution, and shows that alleviating the quadratic attention bottleneck yields tangible wall-clock gains even in an asynchronous, real-hardware setting.

---

### Official Review · Reviewer_HGmx · 2025-10-30

**Soundness:** 3
**Presentation:** 4
**Contribution:** 3
**Rating:** 6
**Confidence:** 3

**Summary:**

This paper proposes Sparse Imagination, a simple yet effective method designed for transformer-based world models. The key idea is to accelerate planning by applying random dropout on visual tokens, significantly improving inference speed while largely preserving predictive performance. Extensive experiments and analyses demonstrate that the random dropout strategy, along with the corresponding attention mechanism used during training, is effective on a variety of robotics tasks.

**Strengths:**

1. The paper introduces Sparse Imagination, a surprisingly simple yet highly effective approach for accelerating world models. Counter-intuitively, such a straightforward method outperforms importance-based sampling strategies, offering both practical value and novel insight that the full patch tokens are redundant.

2. The authors conduct extensive experiments to demonstrate the superiority of random dropout–based Sparse Imagination. Comparisons across multiple dropout ratios and different dropout strategies provide strong empirical support for the central claim.

**Weaknesses:**

1. The real-world evaluation is limited to only a single task, which raises concerns about the robustness of the reported results. In general, I would suggest using at least three tasks, or at a minimum two different embodiments. This is particularly important because the workspace of the LeRobot arm is quite small, meaning that the effective action chunk workspace is very limited.

2. I have some concerns regarding the comparative experimental settings. In the case of random sampling, does fixed imply that the same subset of tokens is dropped consistently across different frames? Concerning the blind spot analysis, the paper suggests that blind spots may originate from static aspects of the scene during initialization. In this context, are the important tokens primarily those associated with the end-effector (EEF)? Moreover, I believe the analysis would be more intuitive and convincing if the authors could provide additional visualizations and concrete examples of blind spots as part of a quality assessment.

3. A minor suggestion: It would be helpful to report an average score across multiple tasks in the experiments. This would make it more intuitive to see the overall superiority of Sparse Imagination compared to other baselines or ablations.

**Questions:**

1. From my understanding of this work, actions are sampled either from MPC or from a VLA policy. The world model then imagines the future states resulting from these actions. My question is: during execution, how are the candidate actions selected based on the imagined states, especially in open-ended robotics tasks where no golden state is available?

2. On line 258, the authors mention that the world model can evaluate and refine trajectories. How exactly is the refinement performed? I could not find a detailed description of this process in the paper.

3. On line 141, the authors mention splitting the visual tokens into two groups. What is the motivation for this design? How are the groups specifically divided — for example, first half vs. second half, interleaved, or one group being dropped out? Why does such a partition help strengthen subtoken-level patterns?

4. To address the issue of full-patch visual tokens being overly redundant for world model learning, there is also the concept of latent action [1] in robotic manipulation, which models the state transition a -> s using only a small number of tokens. I would like to ask how the authors consider the relationship between sparse imagination and latent action. Have you considered making a comparison, since both approaches largely fall within the scope of robotics tasks with action annotations?

References:

[1] Latent Action Pretraining from Videos.

---

> ### Author Response · Authors · 2025-11-21
>
> Thank you for your careful review, positive evaluation of our work, and the constructive feedback that helped us better understand how to strengthen the paper.
>
> **W1. The real-world evaluation is limited to only a single task, which raises concerns about the robustness of the reported results. In general, I would suggest using at least three tasks, or at a minimum two different embodiments. This is particularly important because the workspace of the LeRobot arm is quite small, meaning that the effective action chunk workspace is very limited.**
>
> We agree that evaluating on more real-world tasks would further strengthen the robustness claims. In our current setup, however, we are constrained to a single physical embodiment (the LeRobot SO-101 arm), so expanding to multiple hardware platforms is not feasible within this work.
>
> Within the existing LeRobot setup, we are preparing an additional real-world task with a different objective to further stress-test the method, but this requires a full pipeline of data collection, world-model training, and VLA policy training, so the results are not yet ready. We will report them if completed in time.
>
> **W2. I have some concerns regarding the comparative experimental settings. In the case of random sampling, does fixed imply that the same subset of tokens is dropped consistently across different frames? Concerning the blind spot analysis, the paper suggests that blind spots may originate from static aspects of the scene during initialization. In this context, are the important tokens primarily those associated with the end-effector (EEF)? Moreover, I believe the analysis would be more intuitive and convincing if the authors could provide additional visualizations and concrete examples of blind spots as part of a quality assessment.**
>
> In every methods, for a single iteration, the mask is spatially consistent across all history frames \(O_{t-h:t}\) passed to the world model, i.e., the same spatial locations are dropped in every frame. Also, the “Fixed” sampling mentioned in the comparative experiments (Table 3) means that the token-dropout mask is held constant throughout the entire MPC trajectory optimization process (across all CEM iterations and all candidates) for a given MPC iteration. By contrast, the “Random” strategy samples a new, independent mask at every MPC iteration.
>
> Regarding the blind-spot analysis, your understanding is correct. Static importance metrics tend to assign high scores to tokens that belong to **salient objects such as the robot end-effector (EEF) and the robot body, while repeatedly down-weighting background or less salient regions**. This bias can create persistent blind spots when those neglected regions later become task-relevant and the agent might traverse. You can find this tendency in Figure 6.
>
> To make this more concrete, we add a visualization comparing a static importance-based mask (Attention-Encoder) and Random sampling in PushT (see new Figure 9 in the revised PDF). In this task, a blue ball must push a T-shaped block toward the goal region. The static importance map effectively prunes away part of the workspace based on its early appearance: regions that initially look like “background” are permanently deprioritized, even if they later become traversed and task-relevant. This loss of visual variance makes all candidate actions appear similarly good, collapsing the planning process and often leading to failure. By contrast, the unbiased Random sampling strategy does not systematically ignore any spatial region, so dynamic objects like the blue ball are observed under multiple masks over time, preserving the feature variance needed for robust planning and making such blind-spot failures much less likely. The new figure and added passages are highlighted in blue in the revised PDF for ease of reference.
>
>
> **W3. A minor suggestion: It would be helpful to report an average score across multiple tasks in the experiments. This would make it more intuitive to see the overall superiority of Sparse Imagination compared to other baselines or ablations.**
>
> As suggested, we report averages across the simulated environments in below tables. We believe these changes make the overall performance and efficiency advantages of Sparse Imagination clearer.
>
> **Table 1**
>
> |Drop ratio|Average SR (%)|
> |----|---|
> |Full |70.9|
> |CLS|49.6|
> |10%|74.1|
> |20%|69.6|
> |30%|71.1|
> |40%|68.6|
> |50%|69.7|
> |60%|63.6|
> |70%|53.3|
> |80%|54.3|
> |90%|46.7|
>
> **Table 2**
>
> |Drop ratio|Planning time (s/iter)|Change rate (%)|
> |------|---------|-------|
> |Full |183.3 |–|
> |CLS|71.0|-61.3|
> |10%|166.3 |-9.3 |
> |20%|150.0 |-18.1|
> |30%|136.3 |-25.6|
> |40%|119.8 |-34.7|
> |50%|109.0 |-40.5|
> |60%|99.8|-45.6|
> |70%|89.5|-51.2|
> |80%|81.5|-55.5|
> |90%|73.8|-59.8|
>
> **Table 3**
>
> |Method|Average SR (%)|
> |---------|---------|
> |Random|66.7|
> |Fixed |64.3|
> |LHS |65.7|
> |LTRP|59.5|
> |Attention-Encoder|63.0|
> |STAR|61.4|
> |Attention-WM|64.0|
> |ATC |41.7|

---

> ### Author Response · Authors · 2025-11-21
>
> **Q1. From my understanding of this work, actions are sampled either from MPC or from a VLA policy. The world model then imagines the future states resulting from these actions. My question is: during execution, how are the candidate actions selected based on the imagined states, especially in open-ended robotics tasks where no golden state is available?**
>
> Our methodology in this paper is applied primarily on **goal-conditioned planning**. At test time, candidate action sequences are evaluated by rolling them out through the world model to obtain an imagined future state feature, which is then compared to the goal feature. We select the action sequence that minimizes the feature-space distance, and execute the first action (or action chunk) of this best sequence.
>
> While our experiments are goal-conditioned, the same framework applies more generally as long as a suitable **cost or reward function** can be defined on imagined futures, so it is not limited to settings with an explicit pixel-level “golden state.”
>
> **Q2. On line 258, the authors mention that the world model can evaluate and refine trajectories. How exactly is the refinement performed? I could not find a detailed description of this process in the paper.**
>
> You are right that our original wording was imprecise: it is the **planner** that evaluates trajectories (and, in the CEM-based setting, refines them), while the world model only provides the rollouts used for this evaluation. In the policy-guided variant, by contrast, the planner does not refine trajectories but simply evaluates candidate action sequences proposed by the policy. In the revised manuscript, we therefore changed the sentence on line 258 from “the world model can evaluate and refine trajectories” to “our planner then evaluates these candidates,” and we highlight this edit in blue in the updated PDF.
>
> The “refinement” itself refers to the standard **iterative optimization loop of the Cross-Entropy Method (CEM)** used by the planner. At each planning step, the planner:
>
> 1. Samples $K$ candidate action sequences from a parametric distribution.
>
> 2. Rolls them out through the world model and computes the goal-distance cost for each sequence.
>
> 3. Selects the top $K_{elite}$ sequences as elites.
>
> 4. Updates (refines) the mean and variance of the sampling distribution based on these elites.
>
> This procedure is repeated for several iterations, gradually concentrating the sampling distribution around low-cost trajectories. In this sense, the **planner** uses the world model both to **evaluate** candidate trajectories and to **refine** them via CEM updates guided by imagined rollouts. Again, in our explicit-policy setting (e.g., with a VLA-style policy), we instead sample candidate actions directly from the policy and use the **planner** only to **evaluate** them, without running the CEM refinement loop.
>
> **Q3. On line 141, the authors mention splitting the visual tokens into two groups. What is the motivation for this design? How are the groups specifically divided — for example, first half vs. second half, interleaved, or one group being dropped out? Why does such a partition help strengthen subtoken-level patterns?**
>
> The Randomized Grouped Attention strategy is designed to make the world model **robust to arbitrary token subsets at test time**. Concretely, at each training iteration we:
>
> - Randomly permute the visual tokens and split them into two non-overlapping groups, and
>
> - Apply an attention mask so that tokens only attend to others **within the same group**.
>
> Thus, the grouping is *random* at every iteration (not a fixed first-half/second-half or spatially contiguous split), and no group is permanently dropped. By repeatedly restricting attention to partial contexts in this way, the model is encouraged to learn dynamics that do not rely on always seeing all neighboring patches simultaneously. As a result, at test time the same model can accurately predict future states even when only an arbitrary subset of patches is retained, which is precisely what helps enable Sparse Imagination.

---

> ### Author Response · Authors · 2025-11-21
>
> **Q4. To address the issue of full-patch visual tokens being overly redundant for world model learning, there is also the concept of latent action [1] in robotic manipulation, which models the state transition a -> s using only a small number of tokens. I would like to ask how the authors consider the relationship between sparse imagination and latent action. Have you considered making a comparison, since both approaches largely fall within the scope of robotics tasks with action annotations?**
>
> Thank you for pointing out this connection. We view Sparse Imagination and latent-action methods (such as [1–4]) as **complementary rather than competing** approaches to efficiency.
>
> Latent-action works typically learn a low-dimensional latent action space (often parameterized by a small set of discrete codes) that summarizes the *action-induced transition* between consecutive frames, and then train a policy or world model to operate directly in this compact latent space (representation compression). In contrast, Sparse Imagination is designed specifically for **token-based Transformer world models**, and targets the **spatial redundancy of ViT patch tokens** in the visual encoder and dynamics model by reducing the number of high-dimensional tokens processed during rollouts (computational acceleration).
>
> Because latent-action models usually do not expose a spatial patch grid during policy learning (they operate on short sequences of latent codes rather than hundreds of image patches), our token-level sparsification mechanism does not apply directly, and a fair comparison would require substantial adjustments, which is beyond the scope of this work. We see combining latent-action representations with sparse token processing as an interesting and orthogonal direction for future research.
>
> **References**
>
> [1] Ye, S., Jang, J., Jeon, B., Joo, S., Yang, J., Peng, B., ... & Seo, M. (2024). Latent action pretraining from videos. arXiv preprint arXiv:2410.11758.
>
> [2] Schmidt, D., & Jiang, M. (2023). Learning to act without actions. arXiv preprint arXiv:2312.10812.
>
> [3] Nikulin, A., Zisman, I., Tarasov, D., Lyubaykin, N., Polubarov, A., Kiselev, I., & Kurenkov, V. (2025). Latent action learning requires supervision in the presence of distractors. arXiv preprint arXiv:2502.00379.
>
> [4] Bruce, J., Dennis, M. D., Edwards, A., Parker-Holder, J., Shi, Y., Hughes, E., ... & Rocktäschel, T. (2024). Genie: Generative interactive environments. In Forty-first International Conference on Machine Learning.

---

> ### Comment · Reviewer_HGmx · 2025-11-26
> **Reply to weakness**
>
> Thank you for the authors’ thorough and constructive response, as well as for the additional analyses and visualizations.
>
> **Regarding real-world experiments**. I understand that the number of robot platforms may be constrained by hardware availability. Nevertheless, I believe it would be particularly valuable to design targeted real-robot experiments that specifically probe the blind-spot failure cases. In particular, scenarios where certain sampling strategies systematically ignore task-relevant parts (e.g., object regions that later become important) would further strengthen the empirical evidence. Additional real-world visualizations illustrating such blind-spot phenomena would also be highly informative.
>
> **For the visualization of blind spots**, I find that Figure 9 provides a clear and intuitive illustration of the issue. I do, however, have a few follow-up questions. For example, with a relatively high dropout ratio (e.g., 60%), Random dropout may also occasionally mask out the blue ball in Figure 9. Compared to other sampling strategies, does Random dropout still exhibit fewer or less persistent blind-spot failures in practice? If possible, could the authors provide a rough quantitative comparison or proportion to characterize how often blind spots occur under different sampling methods?
>
> In addition, since the importance-based method appears to rely on DINO-v2 self-attention, does this imply that its performance is, to some extent, limited by the choice of the visual encoder? It would be helpful to better understand whether the observed blind spots are fundamentally tied to the encoder’s attention bias.
>
> Finally, **for the average score results**, I would suggest including these quantitative summaries in the revised version as well, as they can help readers more easily grasp overall performance trends across tasks.

---

> > ### Author Response · Authors · 2025-12-03
> >
> > We sincerely thank the reviewer for carefully revisiting our response and for the constructive suggestions. We have conducted additional analyses and real-world experiments to address these points more concretely. All of your suggestions that we were able to incorporate have been reflected in the revised PDF and are highlighted in blue.
> >
> > ---
> >
> > **Regarding real-world experiments.**
> >
> > We additionally evaluate our method on a new real-robot task, **Put Blue Block in the Drawer and Close it (Drawer)**, alongside the original **PickPlace** setup, and its explanation has been added in **Chapter 4.1, 4.3, and Figure 3**.  For both tasks, we have included visualizations where certain token selection strategies temporarily drop regions that later become important for control; in these cases, the planner has less precise information about the end-effector pose relative to the object than under Random dropout. These examples have been added to the **Appendix D.3 (Figures 23-24)** for the reviewer’s reference.
> >
> > Quantitatively, on the Drawer task, **full-patch planning and 50% dropout** both achieve a **70.0% success rate**, while the **VLA-only** baseline without planning attains **60.0%**; at the same time, the average episode time decreases **from 14.0 s (full) to 10.6 s at 50% dropout**, compared to 8.8 s for the VLA-only baseline. We have added the detailed results in **Figure 5.**
> >
> > ---
> >
> > **For the visualization of blind spots.**
> >
> > To quantify how often blind spots occur, we performed a controlled analysis on the PushT environment during a single planning iteration. At each iteration, CEM samples multiple candidate action sequences, each consisting of 2D displacement actions for the blue ball (agent). Since the environment state already provides the current ball center **$(x_t, y_t)$**, we can compute the ball’s hypothetical future centers under each candidate by simply applying the cumulative 2D displacements.
> >
> > We then project these centers into image space; by converting these coordinates to the image resolution, we can determine which patch each center falls into. We approximate the blue ball as a circle whose diameter equals one patch size and use this circle to determine which patch tokens it overlaps. We count a **blind-spot event** whenever **all** overlapping tokens for the ball are included in the dropped-token set. Aggregating over all candidates and timesteps, we obtain the fraction of agent states that fall into a blind spot for 60 episodes.
> >
> > At a **50% drop ratio**, **Random dropout** produces blind spots in only ≈3% of candidate states, whereas other methods mask the agent in ≈18–27% of states. At **60% drop**, blind-spot rates increase for all methods, but **Random** still remains substantially lower (≈12%) than baselines (≈23–33%). For completeness, the measured blind-spot rates are:
> >
> > | Method    | 50% drop | 60% drop |
> > |----------|---------:|---------:|
> > | Attention-WM | 26.60%   | 31.17%   |
> > | Attention-Encoder     | 20.99%   | 33.04%   |
> > | STAR    | 17.74%   | 22.87%   |
> > | LTRP     | 18.56%   | 27.05%   |
> > | Random   | 3.10%    | 12.26%   |
> >
> > **These results show that, at comparable sparsity levels, Random dropout yields substantially fewer blind-spot events than other selection methods.** We have added this in **Appendix C.9 and Table 13**.
> >
> > ---
> >
> > **Regarding the encoder’s effect on blind spots.**
> >
> > We first clarify that our method and Attn-Enc use the original **DINO [1]**, rather than **DINO-v2 [2]**. We agree that any sampler that ranks tokens based on encoder attention will be influenced by the backbone’s attention bias. Prior works have shown that **self-supervised ViT-style encoders such as DINO tend to train self-attention to capture object-level patterns in later layers**, and that this behavior is **shared across related ViT architectures [1, 3]**. Thus, the type of saliency bias we show is **not specific to a particular type of encoder**.
> >
> > To check whether our blind-spot phenomenon is tied to a specific encoder, we additionally replicated our blind-spot visualizations with two other pretrained ViT-style encoders, **MoCo-v3** and **DINOv3**, using the same Attn-Enc sampling procedure. For all of these encoders, we still observe that **it can create blind spots, whereas Random dropout does not follow any particular attention pattern**. The corresponding visualizations are now included in **Appendix D.3 (Figure 22)**.
> >
> > ---
> >
> > **Average score results.**
> >
> > We agree that presenting average performance summaries improves readability. In the revised version, we have included the average-score results (across tasks) in **Tables 1-3.** We also corrected a typo in Table 1 for the Wall task, where the values for “Full” and “CLS” were inadvertently swapped; the table has been updated accordingly, and the corrected entries are now highlighted in blue.

---

> ### Comment · Reviewer_HGmx · 2025-11-26
> **Reply to questions**
>
> Thank you for the authors’ thoughtful response and the clarification provided. These responses address most of my questions. For Q1 I have additional concerns:
>
> Q1: I understand goal-conditioned planning. However, I am more interested in how the method works in non-goal-conditioned scenarios, such as real-robot experiments, where there is no well-defined end goal state. In these settings, I would like the authors to clarify the execution pipeline—for example, how the imagined rollouts produced by the world model are evaluated in real-world tasks. The authors mention using a reward or cost function, but it is not clear how such a function is designed in practice. I find this description somewhat vague and would appreciate further clarification.

---

> > ### Author Response · Authors · 2025-12-03
> >
> > **References**
> >
> > [1] Caron, M., Touvron, H., Misra, I., Jégou, H., Mairal, J., Bojanowski, P., & Joulin, A. (2021). Emerging properties in self-supervised vision transformers. In Proceedings of the IEEE/CVF international conference on computer vision (pp. 9650-9660).
> >
> > [2] Oquab, M., Darcet, T., Moutakanni, T., Vo, H., Szafraniec, M., Khalidov, V., ... & Bojanowski, P. (2023). Dinov2: Learning robust visual features without supervision. arXiv preprint arXiv:2304.07193.
> >
> > [3] Park, N., Kim, W., Heo, B., Kim, T., & Yun, S. (2023). What do self-supervised vision transformers learn?. arXiv preprint arXiv:2305.00729.

---

> > ### Author Response · Authors · 2025-12-03
> >
> > We thank the reviewer for the helpful follow-up question and for encouraging us to clarify the non-goal-conditioned setting more concretely.
> >
> > ---
> >
> > In **non-goal-conditioned scenarios** without an explicit goal state, a typical setting is where a task is given via **language instructions**. In such settings, one can leverage **pretrained models** to instantiate a goal representation or an objective cost function, and then plug that into the same planning loop. For example, **text-conditioned diffusion models** can generate plausible future goal images consistent with a language description [1], **pretrained VLMs** can answer whether a visual state satisfies a text instruction or how close it is to success [2, 3], and **fine-tuned VLMs** can provide auxiliary progress signals such as “steps-to-completion” [4]. These works offer concrete recipes for constructing success / reward signals from language, which can be combined with imagined rollouts.
> >
> > In our framework, the world model rolls out in only a subset of tokens, and the planner evaluates each imagined future based on this subset. In the goal-conditioned experiments in the paper, we embed both the sparsely imagined future tokens and the goal image using the same vision backbone, and define a **feature-space distance as the cost**, which is minimized by MPC-CEM. For a language-conditioned variant, there are **two natural options**:
> >
> > **(1)** If a text-conditioned diffusion model such as [1] is used to generate **a goal image from the instruction**, it can play the role of the goal image in our current formulation, and the existing goal-conditioned pipeline can be applied **without further modification**.
> >
> > **(2)** Alternatively, one can replace the goal-distance cost with a **reward scorer built from VLMs** like [2, 3, 4]: for each imagined trajectory, the sparsely imagined token subset is passed through a vision (or vision–language) encoder jointly with the instruction text to yield a **score regarding task success**. This score then serves as the trajectory cost inside the same MPC-CEM loop, again **without changing the sparse imagination mechanism itself**.
> >
> > A practical consideration in the VLM setting is that the downstream success scorer must be able to operate on **sparse visual token subsets**. Analogous to how we train the world model with **randomized grouped attention** so that it can reliably imagine from randomly sampled visual tokens, one could lightly fine-tune the success scorer under **random token sub-sampling**, encouraging it to remain stable even when only a subset of visual tokens is available. Although our current experiments focus on visual goals, our method is inherently compatible with these settings, because it only requires either a **goal representation** (e.g., an image embedding) or a **reward / success score** for each imagined trajectory, so integrating such success functions on sparse tokens is conceptually straightforward and a natural direction for future work.
> >
> > ---
> >
> > **References**
> >
> > [1] Du, Y., Yang, S., Dai, B., Dai, H., Nachum, O., Tenenbaum, J., ... & Abbeel, P. (2023). Learning universal policies via text-guided video generation. Advances in neural information processing systems, 36, 9156-9172.
> >
> > [2] Du, Y., Konyushkova, K., Denil, M., Raju, A., Landon, J., Hill, F., ... & Cabi, S. (2023). Vision-language models as success detectors. arXiv preprint arXiv:2303.07280.
> >
> > [3] Chen, B., Xu, Z., Kirmani, S., Ichter, B., Sadigh, D., Guibas, L., & Xia, F. (2024). Spatialvlm: Endowing vision-language models with spatial reasoning capabilities. In Proceedings of the IEEE/CVF Conference on Computer Vision and Pattern Recognition (pp. 14455-14465).
> >
> > [4] Du, Y., Yang, M., Florence, P., Xia, F., Wahid, A., Ichter, B., ... & Tompson, J. (2023). Video language planning. arXiv preprint arXiv:2310.10625.

---

### Official Review · Reviewer_C7kW · 2025-10-31

**Soundness:** 3
**Presentation:** 2
**Contribution:** 3
**Rating:** 4
**Confidence:** 4

**Summary:**

This paper introduces a method for accelerating visual world-model planning by processing only a randomly selected subset of vision tokens during latent rollouts. The approach trains a transformer-based world model with randomized grouped attention so that, at test time, the planner can flexibly reduce token count without modifying the architecture. During planning, a fraction of tokens is randomly retained and used for MPC/CEM rollouts, yielding substantial computational savings with minimal degradation in performance. The work compares against learned token-importance and compression baselines, showing that random token selection matches or outperforms these more complex strategies. Experiments across serveral simulated domains and a real-world robotic task demonstrate that dropping 10–50% of tokens preserves task success while significantly reducing planning time. The authors argue that static importance metrics suffer from “blind spots” in dynamic planning settings, while random sampling avoids such failures with negligible overhead. Overall, the paper positions random token dropout as a simple and practical baseline for efficient visual planning.

**Strengths:**

1. The paper is generally well-written and easy to follow, with a clear problem formulation, motivation, and method description. The overall narrative is coherent and the technical contributions are communicated effectively.

2. The core idea of training a transformer-based world model with randomized grouped attention such that it can perform test-time planning by randomly dropping visual patch tokens is conceptually simple, architecture-agnostic, and broadly applicable to visual world-modeling settings.

3. Experiments across multiple simulated control environments (e.g., PointMaze, Wall, PushT, Rope) demonstrate that dropping approximately 50% of visual tokens yields substantial planning-time reductions (40–60%) while preserving control performance. These results show that the approach delivers consistent and meaningful efficiency gains with minimal performance degradation.

4. The method is compared against learned or attention-based token selection or merging techniques (e.g., LTRP, STAR, ATC), and consistently matches or outperforms them. This provides compelling evidence that simple random sampling can avoid failure modes associated with static importance-scoring heuristics and offers a strong, low-overhead baseline for token reduction in visual planning.

5. Real-robot results and integration with a Vision-Language-Action model (SmolVLA) further strengthen the empirical validation. The technique transfers effectively to real-world robotic control and maintains competitive task success while reducing real-world episode execution time by roughly half, demonstrating practical benefit in realistic deployment scenarios.

**Weaknesses:**

1. The writing, particularly in the Experiments section, needs improvement for readability. Important details and baselines are placed in Appendix B.6 instead of the main text, which makes it difficult to follow comparisons and understand experimental context. Additionally, some methods (e.g., Latin Hypercube Sampling, McKay et al., 2000) are referenced without explanation, making it harder for readers unfamiliar with these techniques to interpret results.

2. The paper argues that training the world model with random patch dropout induces robustness to token sparsification at test time, but this claim is supported primarily by empirical evidence. A simple formal or conceptual analysis clarifying when and why random token subsets preserve planning quality would strengthen the contribution. For instance, discussing conditions related to scene observability, task complexity, visual redundancy, and environmental structure would help clarify the boundaries of the method’s effectiveness. Explicitly connecting these factors to the observed planning success patterns would provide a stronger theoretical grounding and a clearer understanding of the method’s limitations and applicability.

3. The experiments focus on goal-conditioned manipulation tasks, and it is unclear whether the approach extends to broader embodied settings (e.g., DMControl, Meta-World, RLBench, BEHAVIOR-1K, Habitat 3.0, or long-horizon household environments). Likewise, the evaluation uses a narrow set of visual encoders. Given recent advances in visual backbones (e.g., DINOv3, FastVLM, OpenVision-2, SAIL-VL2), testing across a wider set of pretrained representations would strengthen the claim that random patch sparsification generalizes beyond the chosen model family.

4. The proposed approach is evaluated primarily within an MPC/CEM planning framework, with an additional VLA-guided sampling variant for real-robot experiments. However, it remains unclear whether the computational and performance benefits of this method extend to other planning paradigms. In particular, it would be useful to see results under implicit planners used in model-based RL (e.g., actor-critic methods) and learned planning approaches (e.g., transformer-based sequential planners or diffusion-policy rollouts). Because different planners have varying tolerance to approximation errors and representation sparsity, demonstrating consistent benefits would help establish that random token dropout is a generally applicable mechanism rather than one primarily suited to CEM-style sampling-based planning.

5. The method relies on a manually chosen token-drop ratio, tuned per task. While the simplicity of random dropout is appealing, the lack of adaptive control limits its practicality in settings where computation budgets or scene complexity change over time (e.g., dynamic camera views or occlusions). An adaptive mechanism that adjusts sparsity based on model confidence, rollout divergence, uncertainty estimates, or planning consistency could potentially offer better performance–efficiency trade-offs. The paper briefly acknowledges this possibility, but does not explore even lightweight heuristics (e.g., annealing token sparsity or entropy-based token retention). As a result, it is unclear whether token sparsity could be automatically optimized online, which would be important for real-time deployment and applications with fluctuating computational constraints.

6. Although the reported runtime savings are substantial, the compute analysis remains relatively coarse-grained. The paper primarily reports end-to-end planning time, but does not clearly decompose where efficiency gains arise (e.g., savings in vision attention layers, latent rollout computation, sampling loops, and planning iterations). Providing latency per module across token-drop ratios would help understand where the speedups come from and how to tune the method for different hardware and latency budgets.

**Questions:**

1. How sensitive is the approach to the choice of token-drop ratio across environments? Did you observe cases where higher or lower sparsity meaningfully changed success rates or planning stability?

2. Can the authors formalize or provide intuition for the conditions under which random token subsets preserve planning quality? For example, how do factors like scene observability, task complexity, and visual redundancy influence performance?

3. Do you expect similar efficiency gains on non-manipulation tasks such as navigation, locomotion, or long-horizon household planning? Have you tested (or can you comment on) performance in these settings?

4. Would this method apply to other planning paradigms beyond CEM-based MPC, such as actor-critic agents, or transformer-based learned planners?

5. Can you consider providing some insight into adaptive mechanisms for adjusting the token-drop ratio at test time (e.g., confidence- or uncertainty-based heuristics)? Do you foresee a principled approach for online adaptation?

6. Can the authors provide a breakdown of computation cost across components (e.g., attention layers, latent rollouts, sampling iterations, VLA calls) to clarify where efficiency gains originate?

---

> ### Author Response · Authors · 2025-11-21
>
> We thank the reviewer for the careful and constructive review, and in particular for highlighting both the practical strengths of our random token dropout approach and the concrete directions to clarify its analysis, scope, and experimental presentation.
>
> **W1. The writing, particularly in the Experiments section, needs improvement for readability. Important details and baselines are placed in Appendix B.6 instead of the main text, which makes it difficult to follow comparisons and understand experimental context. Additionally, some methods (e.g., Latin Hypercube Sampling, McKay et al., 2000) are referenced without explanation, making it harder for readers unfamiliar with these techniques to interpret results.**
>
> Thank you for pointing this out. In the revised manuscript, we have made Sec. 4.4 more self-contained by introducing a dedicated subsection, **“Token Reduction Methods,”** where we (i) group all baselines into four families, Random Sampling (Random, Fixed, LHS), Learning-Based Pruning (LTRP), Attention-Based Pruning (Attention-Encoder, STAR, Attention-WM), and Cluster & Merging (ATC), and (ii) provide a concise description of how each method selects or merges tokens. In particular, we now explicitly explain Latin Hypercube Sampling (LHS) as a stratified uniform sampling scheme over the 2D patch grid in the manuscript.
>
> We also clarify that all token-reduction baselines share the same world model and planning hyperparameters and retain the same number of tokens, ensuring directly comparable conditions. We believe these changes substantially improve the readability and interpretability of our experimental comparisons. **Newly added passages related to this point are highlighted in blue in the revised PDF (Section 4.4, Appendix B.6).**
>
> **W2. The paper argues that training the world model with random patch dropout induces robustness to token sparsification at test time, but this claim is supported primarily by empirical evidence. A simple formal or conceptual analysis clarifying when and why random token subsets preserve planning quality would strengthen the contribution. For instance, discussing conditions related to scene observability, task complexity, visual redundancy, and environmental structure would help clarify the boundaries of the method’s effectiveness. Explicitly connecting these factors to the observed planning success patterns would provide a stronger theoretical grounding and a clearer understanding of the method’s limitations and applicability.**
>
> We agree that performance is influenced by factors such as visual redundancy and task difficulty. However, in standard benchmarks these factors tend to be entangled rather than independently controllable, which makes a fully disentangled formal analysis challenging in practice. Instead, we study them empirically using simple proxies and cross-environment comparisons, and plausibly explain the observed success patterns using **visual redundancy and task difficulty.**
>
> **(i) Visual redundancy.**
>
> We compare Wall and PointMaze, which are both 2D navigation tasks with similar control structure but markedly different visual textures: Wall has a simple, mostly uniform background, whereas PointMaze uses a high-frequency checkerboard pattern. At a 90% drop ratio, Wall still achieves 86.7% success, while PointMaze drops to 71.7%. One plausible interpretation is that in visually simpler scenes, similar information is more redundantly distributed across patches, so planning can tolerate more aggressive sparsification,
> whereas in highly textured scenes the same drop ratio more readily degrades the performance.
>
> **(ii) Task difficulty**
>
> Wall and PushT share relatively clean, top-down visuals (similar scene complexity), but Wall is a relatively simple navigation task whereas PushT involves more challenging, fine-grained manipulation. Wall tolerates very high sparsity and retains near 90% success even at large drop ratios, whereas PushT begins to degrade more noticeably beyond 50% dropout. This suggests that more difficult tasks that depend on subtle geometric details require a higher effective token density.
>
> These explanations do not imply that visual redundancy and task difficulty are the only factors at play; other aspects like scene observability may also affect sensitivity to sparsification. **Our goal here is not to provide a complete theory, but to add intuitive structure on top of the experimental results, clarifying when sparse imagination worked well in our benchmarks and where it started to break down.**

---

> ### Author Response · Authors · 2025-11-21
>
> **W3. The experiments focus on goal-conditioned manipulation tasks, and it is unclear whether the approach extends to broader embodied settings (e.g., DMControl, Meta-World, RLBench, BEHAVIOR-1K, Habitat 3.0, or long-horizon household environments). Likewise, the evaluation uses a narrow set of visual encoders. Given recent advances in visual backbones (e.g., DINOv3, FastVLM, OpenVision-2, SAIL-VL2), testing across a wider set of pretrained representations would strengthen the claim that random patch sparsification generalizes beyond the chosen model family.**
>
> Our method is designed to exploit redundancy in ViT-style patch-token representations and does not rely on a specific encoder architecture. In the current submission, we already evaluate separate world models built on top of DINO, MoCo-v3, and MAE encoders. To test a more recent backbone, we additionally ran an experiment (PointMaze, 60 episodes, same setting with MoCo-v3, DINO, and MAE) with DINOv3 (ViT-S/16) [1] and observed the same behavior: planning with a 50\% token drop achieved a 98.3\% success rate, matching the Full-Patch baseline (98.3\%). This supports our claim that random patch sparsification is not tied to a specific visual encoder and can transfer to stronger, state-of-the-art ViT-based backbones.
>
> Although our experiments are formulated as goal-conditioned tasks, the framework itself is not restricted to this setting: as long as a suitable cost or reward can be defined on imagined futures, the same sparse rollouts can be used to evaluate and compare candidate trajectories.
>
> Regarding broader embodied settings, our experiments already cover a mix of navigation and manipulation-style tasks, including complex scenarios such as LIBERO-10 and our real-world LeRobot setup, where the agent must execute longer-horizon behaviors. We agree that extending the evaluation to additional suites such as DMControl, Meta-World, RLBench, or Habitat 3.0 would further strengthen the generality claim. As a first step in this direction, we are currently porting our pipeline to Meta-World and running experiments on a subset of tasks; these runs are still in progress at the time of this response. If the experiments complete in time, we will include a brief summary of the Meta-World results.
>
> **W4. The proposed approach is evaluated primarily within an MPC/CEM planning framework, with an additional VLA-guided sampling variant for real-robot experiments. However, it remains unclear whether the computational and performance benefits of this method extend to other planning paradigms. In particular, it would be useful to see results under implicit planners used in model-based RL (e.g., actor-critic methods) and learned planning approaches (e.g., transformer-based sequential planners or diffusion-policy rollouts). Because different planners have varying tolerance to approximation errors and representation sparsity, demonstrating consistent benefits would help establish that random token dropout is a generally applicable mechanism rather than one primarily suited to CEM-style sampling-based planning.**
>
> Our method is not tied to a specific planning algorithm: it only assumes (i) a set of visual tokens processed by attention in the world model, and (ii) the ability to randomly subsample these tokens during training and inference. Any planner that queries such a world model can benefit from reduced attention cost during latent rollouts. In this sense, we view patch-token Transformer world models and CEM-style MPC as a representative testbed rather than the only viable application.
>
> To directly test whether our sparsification scheme is compatible with a different planner, we additionally evaluated a gradient-descent-based planner (MPC-GD) instead of using CEM. In both PointMaze and Wall (average success rates of 50 episodes), sparse imagination with a 50% drop ratio achieved essentially the same success rates as the full-patch model, suggesting that random token sparsification preserves the optimization landscape for gradient-based MPC as well. Full results are presented in the below table.
>
> | Drop Ratio   |   PointMaze |   Wall |
> |:-------------|-------:|-------:|
> | Full         |   0.94 |   0.88 |
> | 10%          |   0.94 |   0.9  |
> | 30%          |   0.98 |   0.92 |
> | 50%          |   0.94 |   0.88 |
> | 70%          |   0.9  |   0.82 |
> | 90%          |   0.78 |   0.78 |
>
> A systematic evaluation with implicit planners (e.g., actor-critic or learned planning architectures) is an interesting direction for future work, and we believe our sparsification mechanism can be applied there as long as they rely on transformers.

---

> ### Author Response · Authors · 2025-11-21
>
> **W5. The method relies on a manually chosen token-drop ratio, tuned per task. While the simplicity of random dropout is appealing, the lack of adaptive control limits its practicality in settings where computation budgets or scene complexity change over time (e.g., dynamic camera views or occlusions). An adaptive mechanism that adjusts sparsity based on model confidence, rollout divergence, uncertainty estimates, or planning consistency could potentially offer better performance–efficiency trade-offs. The paper briefly acknowledges this possibility, but does not explore even lightweight heuristics (e.g., annealing token sparsity or entropy-based token retention). As a result, it is unclear whether token sparsity could be automatically optimized online, which would be important for real-time deployment and applications with fluctuating computational constraints.**
>
> In line with the reviewer’s suggestion, we implemented a simple uncertainty-aware adaptive planner. The core idea is to adjust the token drop ratio based on the variance of latent rollouts across MPC samples, using this variance as a proxy for prediction uncertainty: when uncertainty increases between iterations, the planner automatically reduces the drop ratio (i.e., keeps more tokens), and when uncertainty is low, it can safely increase drop ratio.
>
> For the adaptive mechanism, we evaluated two uncertainty-aware variants, **Ada-Rand** and **Ada-Inc**.
> In **Ada-Rand**, at each MPC iteration we (i) compute the variance of latent rollouts across elite trajectories, (ii) compare it to the previous iteration, and (iii) update the drop ratio with this change in variance (higher uncertainty → lower drop ratio, lower uncertainty → higher drop ratio). We then sample a new mask according to the updated drop ratio
>
> In **Ada-Inc**, we use the same variance signal but update the mask *incrementally* rather than resampling from scratch. Starting from the previous mask, tokens with persistently low variance across elites are gradually dropped, while tokens whose latent features exhibit high variance are preferentially added or retained. When the variance signal indicates that the keep ratio should increase (i.e., we need to reduce sparsity), we randomly revive tokens from the previously dropped set and bring them back into the mask, so that sparsity can be adjusted without permanently excluding any region.
>
> In the Wall environment, **Ada-Rand** achieved a 95.0% success rate with an average drop ratio of 26.1%, matching the performance of a strong fixed 50%–drop baseline, while **Ada-Inc** achieved 91.7% success at 29.8% average drop. In PushT, **Ada-Rand** reached 68.3% success (29.0% average drop), and **Ada-Inc** reached 65.0% (35.8% average drop). They match the performance of the best fixed drop ratios (e.g., 10% or 50% fixed drop), and they avoid the sharp performance collapses observed for overly aggressive fixed sparsity.
>
> **Our goal here is not to claim that a particular fixed or adaptive schedule is universally optimal, but to show that our framework readily supports lightweight, uncertainty-based adaptation without additional training.** In particular, because the world model is explicitly trained to be robust under random sparsification (via randomized grouped attention), **the same Sparse Imagination mechanism can be reused at test time to adjust the drop ratio online rather than redesigning the planner or retraining the model.** The fixed-ratio results characterize how performance behaves as a function of sparsity (robust up to around 50\% in our benchmarks), and the adaptive variants illustrate that the same mechanism can be applied when the optimal drop ratio is unknown or varying, which we see as a promising direction for future work.

---

> ### Author Response · Authors · 2025-11-21
>
> **W6. Although the reported runtime savings are substantial, the compute analysis remains relatively coarse-grained. The paper primarily reports end-to-end planning time, but does not clearly decompose where efficiency gains arise (e.g., savings in vision attention layers, latent rollout computation, sampling loops, and planning iterations). Providing latency per module across token-drop ratios would help understand where the speedups come from and how to tune the method for different hardware and latency budgets.**
>
> We agree that a more fine-grained compute analysis helps clarify where the speedups come from. To this end, we profiled the latency of each major component in our planning loop across different token counts in PushT and PointMaze. The decomposition in below tables shows that the dominant acceleration comes from the world model’s self-attention layers during latent rollouts, while other modules either scale linearly with the number of tokens or remain approximately constant.
>
> Concretely, in PushT, the world model self-attention core (the main $O(N^2)$ term) closely follows the expected quadratic scaling: reducing the number of visual tokens from 196 to 98 (50% drop) reduces self-attention time from 87.4 s to 25.1 s (−71%), which is close to the 75% reduction predicted by quadratic scaling. The QKV projections and feed-forward (MLP) layers scale approximately linearly with the token count (about 50% reduction at 50% sparsity), while the image encoder, environment interaction step, and other overheads remain nearly constant across token-drop ratios.
>
> Table 1: Latency breakdown per planning iteration on PushT across different token counts.
> All values are in seconds; “Rel.” columns are relative to the full-token (drop ratio 0%) setting.
>
> | # Drop ratio (%) | WM self-attn core (s) | Self-attn rel. (%) | WM proj (QKV) (s) | WM MLP (FFN) (s) | QKV rel. (%) | MLP rel. (%) | Image encoder (s) | Env / sim step (s) | Other overhead (s) | Total per iteration (s) |
> |:-----:|----------------------:|--------------------:|------------------:|-----------------:|-------------:|-------------:|------------------:|-------------------:|-------------------:|------------------------:|
> | 0 | 87.4 | 100.0 | 30.5 | 34.5 | 100.0 | 100.0 | 11.9 | 2.8 | 10.8 | 177.8 |
> | 10 | 73.3 | 84.0 | 27.5 | 31.1 | 90.1 | 90.3 | 11.7 | 3.4 | 10.3 | 157.4 |
> | 20 | 57.2 | 65.4 | 24.6 | 27.9 | 80.8 | 81.0 | 11.9 | 4.7 | 11.1 | 137.4 |
> | 30 | 47.0 | 53.8 | 21.6 | 24.5 | 70.8 | 71.2 | 11.9 | 3.4 | 10.6 | 119.0 |
> | 40 | 32.9 | 37.7 | 18.7 | 21.1 | 61.3 | 61.3 | 11.8 | 3.5 | 11.6 | 99.8 |
> | 50 | 25.1 | 28.8 | 15.5 | 17.5 | 50.7 | 50.8 | 11.8 | 2.8 | 10.3 | 83.0 |
> | 60 | 16.0 | 18.3 | 12.5 | 14.0 | 40.8 | 40.7 | 11.8 | 4.0 | 10.5 | 68.7 |
> | 70 | 10.6 | 12.1 | 9.5 | 10.7 | 31.2 | 30.9 | 11.8 | 2.4 | 11.1 | 56.1 |
> | 80 | 5.0 | 5.7 | 6.4 | 7.1 | 20.8 | 20.7 | 11.8 | 3.3 | 10.4 | 44.0 |
> | 90 | 2.4 | 2.8 | 3.5 | 3.8 | 11.5 | 11.1 | 11.7 | 3.4 | 10.3 | 35.1 |
>
> Table 2: Latency breakdown per planning iteration on PointMaze across different token counts.
> All values are in seconds; “Rel.” columns are relative to the full-token (drop ratio 0%) setting.
>
> | Drop ratio (%) | WM self-attn core (s) | Self-attn rel. (%) | WM proj (QKV) (s) | WM MLP (FFN) (s) | QKV rel. (%) | MLP rel. (%) | Image encoder (s) | Env / sim step (s) | Other overhead (s) | Total per iteration (s) |
> |:---------------:|----------------------:|--------------------:|------------------:|-----------------:|-------------:|-------------:|------------------:|-------------------:|-------------------:|------------------------:|
> | 0 | 97.0 | 100.0 | 34.7 | 39.1 | 100.0 | 100.0 | 13.3 | 2.8 | 9.9 | 196.7 |
> | 10 | 81.5 | 84.0 | 31.3 | 35.3 | 90.1 | 90.4 | 13.3 | 2.8 | 10.7 | 174.9 |
> | 20 | 63.7 | 65.6 | 28.2 | 31.8 | 81.2 | 81.4 | 13.3 | 2.7 | 10.2 | 149.9 |
> | 30 | 52.3 | 53.9 | 24.5 | 27.8 | 70.7 | 71.2 | 13.5 | 3.6 | 10.4 | 132.1 |
> | 40 | 36.7 | 37.8 | 21.4 | 24.1 | 61.5 | 61.6 | 13.2 | 2.7 | 10.2 | 108.3 |
> | 50 | 28.1 | 29.0 | 17.8 | 20.0 | 51.2 | 51.3 | 13.2 | 3.9 | 9.8 | 92.8 |
> | 60 | 17.8 | 18.4 | 14.2 | 16.0 | 40.9 | 40.8 | 13.3 | 3.4 | 9.6 | 74.4 |
> | 70 | 11.8 | 12.2 | 10.8 | 12.1 | 31.2 | 31.0 | 13.1 | 2.7 | 9.4 | 60.0 |
> | 80 | 5.5 | 5.7 | 7.2 | 8.1 | 20.8 | 20.6 | 13.1 | 2.6 | 10.6 | 47.1 |
> | 90 | 2.7 | 2.8 | 3.9 | 4.3 | 11.2 | 11.0 | 12.9 | 3.4 | 9.5 | 36.7 |

---

> ### Author Response · Authors · 2025-11-21
>
> **Q1. How sensitive is the approach to the choice of token-drop ratio across environments? Did you observe cases where higher or lower sparsity meaningfully changed success rates or planning stability?**
>
> Overall, we observe that the method is relatively insensitive to the exact drop ratio up to about 50% across all environments in Table 1, and even harder environments such as LIBERO and LeRobot: success rates in this range are typically very close to the Full-Patch baseline, and we did not see noticeable instabilities in planning.
>
> Beyond 50%, the sensitivity becomes environment-dependent. Tasks with visually simple and redundant scenes and coarse objectives (e.g., Wall) remain robust even under very high sparsity, retaining high success rates at 80–90% dropout. In contrast, tasks that require fine-grained spatial reasoning and contact (e.g., PushT) exhibit a clearer performance decline once the dropout exceeds roughly 60–70%.
>
> These empirical patterns are in line with our analysis in W2: they suggest that visual redundancy and task difficulty could explain how far sparsification can be pushed without harming performance. In standard benchmarks, however, these factors are often entangled rather than independently controllable, which makes a fully disentangled formal analysis challenging in practice.
>
> **Q2. Can the authors formalize or provide intuition for the conditions under which random token subsets preserve planning quality? For example, how do factors like scene observability, task complexity, and visual redundancy influence performance?**
>
> Please see our response to W2.
>
> **Q3. Do you expect similar efficiency gains on non-manipulation tasks such as navigation, locomotion, or long-horizon household planning? Have you tested (or can you comment on) performance in these settings?**
>
> We expect the efficiency gains to transfer broadly because the main savings come from a structural property of the architecture rather than from a specific task: reducing the number of visual tokens directly alleviates the $O(N^2)$ self-attention cost in the world model. Our current evaluation already includes non-manipulation domains, since PointMaze and Wall are navigation tasks, and we observe the similar qualitative pattern there as in manipulation: planning remains robust up to roughly a 50% drop ratio, with environment-dependent degradation only at higher sparsity levels. Empirically, our results suggest that sensitivity is driven less by the high-level task category (navigation vs. manipulation) and more by visual redundancy and task difficulty.
>
> **Q4. Would this method apply to other planning paradigms beyond CEM-based MPC, such as actor-critic agents, or transformer-based learned planners?**
>
> Please see our response to W4.
>
> **Q5. Can you consider providing some insight into adaptive mechanisms for adjusting the token-drop ratio at test time (e.g., confidence- or uncertainty-based heuristics)? Do you foresee a principled approach for online adaptation?**
>
> Please see our response to W5.
>
> **Q6. Can the authors provide a breakdown of computation cost across components (e.g., attention layers, latent rollouts, sampling iterations, VLA calls) to clarify where efficiency gains originate?**
>
> Please see our response to W6.
>
> **References**
>
> [1] Siméoni, O., Vo, H. V., Seitzer, M., Baldassarre, F., Oquab, M., Jose, C., ... & Bojanowski, P. (2025). Dinov3. arXiv preprint arXiv:2508.10104.

---

> ### Author Response · Authors · 2025-12-03
>
> **Additional Meta-World and Real-World Experiments.**
>
> To address the reviewer’s suggestion regarding broader embodied benchmarks, we evaluated our method to the Meta-World [1] environment. We use SmolVLA to sample 10 action chunk candidates for MPC planning and evaluate them on 50 tasks with 15 trials each. We presented success rates across task difficulties and planning time (per episode) in Chapter 4.3 and Table 3 in the revised version.
>
> | Method    | All | Easy  | Medium  | Hard  | Very Hard | Planning Time
> |----------|---------:|---------:|-------:|-------:|-------:|------:|
> | VLA-only | 41.87% |60.95%|  20.61%|  17.78%| 10.67%| 1.95s
> | Full   | 48.80% |66.90% | 29.09% | 28.89% | 14.67% | 3.63s
> | Drop 50% |  47.73% | 65.24% | 27.27% | 33.33% | 12.00% |2.37s
>
> We also additionally evaluated our method on a new real-robot task, Put Blue Block in the Drawer and Close it (Drawer), alongside the original PickPlace setup, and its explanation has been added in Chapter 4.1, 4.3, Figure 3 and Figure 5.
>
> These results show that our 50% token drop approach recovers nearly all of Full-Patch’s success outperforming VLA-only approach while reducing planning computation, aligning with our previous experiments. Our new results from the Meta-World and real-world LeRobot drawer task supports the method’s broader applicability in various settings.
>
> [1] Yu, T., Quillen, D., He, Z., Julian, R., Hausman, K., Finn, C., & Levine, S. (2020, May). Meta-world: A benchmark and evaluation for multi-task and meta reinforcement learning. In Conference on robot learning (pp. 1094-1100). PMLR.

---

### Official Review · Reviewer_ASzo · 2025-11-02

**Soundness:** 2
**Presentation:** 3
**Contribution:** 2
**Rating:** 6
**Confidence:** 5

**Summary:**

This paper introduces a method called sparse imagination, which aims to increase the efficiency for test-time planning with patch token-based transformer world models while maintaining planning performance. The method works by randomly dropping half of the patch tokens when calculating attention, ensuring robust predictions on any subset of tokens. During MPC planning, an arbitrary percentage of tokens can be dropped in each iteration to accelerate world model rollouts. Experiments on seven simulated environments and one real-world environment show that the method achieves comparable planning performance to the full-patch world model, with significantly reduced planning time.

**Strengths:**

- The idea is well-motivated, as planning time and inference cost are major concerns for patch token-based world models that encode rich information about the environment.
- The reduction in inference and planning time is significant, while achieving comparable or even better planning success rates across various benchmarks.
- The analysis of token information in Section 5.3 is quite interesting, providing insights into the information content and redundancy of patch features.

**Weaknesses:**

- The application of the proposed method seems somewhat limited, as it only applies to world models with patch tokens, a transformer backbone, and MPC as the planning algorithm. However, the idea of randomly dropping tokens during training and inference appears more general. Could this approach be extended to other use cases?
- At planning time, the method relies on resampling different tokens across MPC iterations to capture the full task information. For open-loop CEM planning, however, information is lost since token sampling occurs only once. How are the CEM-only results in Table 1 affected by this limitation?
- The paper could be strengthened by evaluating patch token-based world models using features other than DINO-v2. Does the analysis of patch token information in Section 5.3 generalize to other pre-trained patch features?
- There is no evaluation of the world model’s prediction quality when using full patches while training with random token dropout. Does this training mechanism improve or degrade the world model’s quality?

**Questions:**

- In Table 1, why does a nonzero drop ratio sometimes outperform planning with full patches? Why isn’t the success rate strictly decreasing as the drop ratio increases?
- For the LIBERO and LeRobot tasks, the dataset used to train the world model appears to consist solely of expert trajectories, which provides limited state and action space coverage. How does the world model generate the counterfactual information essential for planning with the policy?
- How does the token drop ratio during training affect the world model’s prediction quality?
-  In Table 2, what does the change in time for CLS mean, given that no tokens are dropped?

---

> ### Author Response · Authors · 2025-11-21
>
> We thank the reviewer for the thoughtful and thorough evaluation of our work and for the constructive comments, which helped us clarify the scope, limitations, and broader implications of sparse imagination.
>
> **W1. The application of the proposed method seems somewhat limited, as it only applies to world models with patch tokens, a transformer backbone, and MPC as the planning algorithm. However, the idea of randomly dropping tokens during training and inference appears more general. Could this approach be extended to other use cases?**
>
> Our method intentionally targets patch-token Transformer world models as a representative testbed, but the underlying mechanism is more general: it only assumes (i) a set of visual tokens processed by self-attention, and (ii) the ability to randomly subsample these tokens during training and inference. In principle, the same sparsification scheme could be applied to other patch-token Transformer architectures, such as video generation or diffusion-based world models, where attention is also the main computational bottleneck. A systematic exploration of these broader applications is beyond the scope of the current submission, but we view our work as a proof-of-concept in this direction.
>
> Regarding the concern that our method may be tied to a particular planning algorithm (e.g., CEM-based MPC), we also tested a gradient-descent-based trajectory optimizer (MPC-GD). In both PointMaze and Wall, sparse imagination with a 50% drop ratio achieved essentially the same success rates as the full-patch model (e.g., 94% vs. 94% in Maze, 88% vs. 88% in Wall), indicating that our random token sparsification is compatible with other planning methods. More detailed success rate results are presented in the below table.
>
> | Drop Ratio   |   PointMaze |   Walll |
> |:-------------|-------:|-------:|
> | Full         |   0.94 |   0.88 |
> | 10%          |   0.94 |   0.9  |
> | 30%          |   0.98 |   0.92 |
> | 50%          |   0.94 |   0.88 |
> | 70%          |   0.9  |   0.82 |
> | 90%          |   0.78 |   0.78 |
>
> **W2. At planning time, the method relies on resampling different tokens across MPC iterations to capture the full task information. For open-loop CEM planning, however, information is lost since token sampling occurs only once. How are the CEM-only results in Table 1 affected by this limitation?**
>
> In the CEM-only settings of Table 1, each episode uses a single randomly sampled token mask that is held fixed throughout planning. Even so, planning performance remains high, indicating that a single sparse subset of tokens per episode already carries sufficient information for the tasks we consider. Also,“Fixed” in Table 3 indicates that using fixed masks through the planning process still achieve strong success rates.
>
> While these results show that fixed masks are viable in practice, we advocate Random dynamic sampling as the default choice for general MPC scenarios: it preserves the simplicity and efficiency of uniform sampling, typically matches or slightly improves over fixed masks, and its unbiased resampling across iterations and episodes reduces the risk of persistent “blind spots”.
>
> **W3. The paper could be strengthened by evaluating patch token-based world models using features other than DINO-v2. Does the analysis of patch token information in Section 5.3 generalize to other pre-trained patch features?**
>
> We clarify that we use DINO ViT-based patch features rather than DINO-v2. To test whether the Section 5.3 token-level analysis depends on this encoder, we repeated the HSIC and attentive probing experiments with an MAE ViT-B backbone. With MAE, Random dropping yields mostly higher HSIC scores and comparable validation loss than attention-based selection across dropout rates, suggesting that random sampling preserves more task-relevant state information in the features.
>
>
> | Drop Ratio | Ours (Random) | Attn-Enc |
> |-----------|----------------|----------------------|
> | 0.0 (Full) | 0.5313 | |
> | CLS | 0.1376 | |
> | 0.01 | 0.5308 | 0.5249 |
> | 0.02 | 0.5307 | 0.5163 |
> | 0.05 | 0.5286 | 0.5149 |
> | 0.1 | 0.5262 | 0.4792 |
> | 0.2 | 0.5221 | 0.4697 |
> | 0.3 | 0.5147 | 0.4246 |
> | 0.4 | 0.5079 | 0.4025 |
> | 0.5 | 0.4977 | 0.3911 |
> | 0.6 | 0.4783 | 0.3852 |
> | 0.7 | 0.4578 | 0.3984 |
> | 0.8 | 0.4188 | 0.3899 |
> | 0.9 | 0.3363 | 0.2900 |
> | 0.95 | 0.2379 | 0.1740 |
> | 0.98 | 0.1617 | 0.1342 |
> | 0.99 | 0.1464 | 0.1250 |
>
> | Drop Ratio | Ours | Attn-Enc |
> |-----------|---------------------|-----------------------|
> | 0.0 (Full) | 0.0518 | |
> | CLS | 0.1449 | |
> | 0.1 | 0.0541±0.0122 | 0.0533 |
> | 0.2 | 0.0709±0.0152 | 0.0627 |
> | 0.3 | 0.0642±0.0104 | 0.0790 |
> | 0.4 | 0.1002±0.0128 | 0.1166 |
> | 0.5 | 0.1018±0.0108 | 0.1260 |
> | 0.6 | 0.1015±0.0117 | 0.1259 |
> | 0.7 | 0.1099±0.0079 | 0.1250 |
> | 0.8 | 0.0999±0.0111 | 0.1131 |
> | 0.9 | 0.1261±0.0230 | 0.1426 |
> | Leave 1 | 0.1284±0.0196 | 0.1484 |

---

> ### Author Response · Authors · 2025-11-21
>
> **W4. There is no evaluation of the world model’s prediction quality when using full patches while training with random token dropout. Does this training mechanism improve or degrade the world model’s quality?**
>
> To assess the world model’s prediction quality, we trained the world models with the optional decoder and measured image reconstruction quality of the predicted features with **LPIPS** [1] **(lower is better)**. The table below compares the LPIPS of the standard Full (full attention) baseline against our model trained with Randomized Grouped Attention (“Drop”) while still decoding from full patches at evaluation time:
>
> | | Full | Drop |
> |-----------|---------------------|-----------------------|
> | **PointMaze** | 0.00028 | 0.00031 |
> | **Wall** | 0.00074 | 0.00078 |
> | **Granular** | 0.05607 | 0.04739 |
> | **Rope** | 0.03347 | 0.01906 |
> | **Pusht** | 0.00561 | 0.00644 |
>
> Across environments, the LPIPS values for the Drop remain very close to the Full-Patch baseline. **These results indicate that our training mechanism does not degrade the world model’s prediction quality under full-patch evaluation while enabling robust performance under sparse imagination at planning time.**
>
> **Q1. In Table 1, why does a nonzero drop ratio sometimes outperform planning with full patches? Why isn’t the success rate strictly decreasing as the drop ratio increases?**
>
> While we do not have a definitive proof, we hypothesize that this counter-intuitive result can be explained by the effects of **stochastic regularization during planning.**
>
> A potential issue with the **Full-Patch** is that it must contend with a high-dimensional and highly redundant state representation. This could cause the planner to converge on a brittle plan that is overly sensitive to spurious visual details or noise in the observation. Such a plan might seem optimal for the current state but could lack the robustness to generalize well during execution.
>
> We believe that moderate random dropout could act as a regularizer to mitigate this potential issue. By introducing stochasticity, it may prevent the planner from becoming overly reliant on any specific set of features and instead encourage it to find more robust solutions that are effective across a variety of slightly perturbed state representations. It is also possible that this regularization effect helps mitigate the **feature oversmoothing** phenomenon, where the representations of an excessive number of similar patches can become indistinguishable, causing a loss of fine-grained detail.
>
> This hypothesis is consistent with our broader finding that task-relevant information is widely distributed across the visual tokens. Because the information is redundant and distributed, removing a small fraction of tokens might not discard critical information. Instead, its primary benefit could be the reduction of noise and the regularization of the planning process, which could explain why a 10-30% dropout may lead to a performance improvement.
>
> **Q2. For the LIBERO and LeRobot tasks, the dataset used to train the world model appears to consist solely of expert trajectories, which provides limited state and action space coverage. How does the world model generate the counterfactual information essential for planning with the policy?**
>
> While the world model for these tasks (LIBERO, LeRobot) is indeed trained exclusively on expert trajectories, our planning setup is designed so that it does not rely on arbitrarily far counterfactuals. In particular, unlike standard CEM or MPC-CEM schemes that sample actions freely and may frequently query the world model with out-of-distribution actions, **the planner for these tasks uses a VLA-policy-guided proposal mechanism**. **Candidate action chunks are generated by a pre-trained VLA policy that imitates expert demonstrations, and the planner only selects among these proposals.**
>
> As a result, most of the rollouts seen by the world model during planning remain close to the support of the expert data distribution: the planner composes and reorders expert-like action segments, rather than exploring completely unconstrained actions. In this regime, **the world model only needs to predict within a relatively local neighborhood around the expert manifold, rather than extrapolating to entirely unseen state-action pairs.**
>
> **Q3. How does the token drop ratio during training affect the world model’s prediction quality?**
>
> To clarify, we do not train with a fixed token drop ratio. Instead, as described in Section 3.1, we use **Randomized Grouped Attention**: at each iteration, tokens are randomly partitioned into groups and attention is restricted within each group, encouraging robustness to arbitrary sparse subsets rather than a specific sparsity level. At test time, the same model can still be run with full-patch attention when desired.
>
> Regarding how the Randomized Grouped Attention strategy affects the world model’s prediction quality, please see our response for W4.

---

> ### Author Response · Authors · 2025-11-21
>
> **Q4. In Table 2, what does the change in time for CLS mean, given that no tokens are dropped?**
>
> In the CLS setting, the world model does not process the patch tokens. Instead, it uses only a single global vector (one token) that summarizes the entire image, analogous to the CLS token in ViT. Since the computational cost of the Transformer world model scales roughly quadratically with the number of input tokens, reducing the input from N spatial patches to a single CLS token leads to a large reduction in computation per imagination step. The time decrease reported for CLS in Table 2 (e.g., −73.4%) therefore reflects the efficiency gain of operating on a single-vector representation. Please note that except CLS setting, every approach does not use the CLS token, they only use patch tokens.
>
> **References**
>
> [1] Zhang, R., Isola, P., Efros, A. A., Shechtman, E., & Wang, O. (2018). The unreasonable effectiveness of deep features as a perceptual metric. In Proceedings of the IEEE conference on computer vision and pattern recognition (pp. 586-595).

---

### Author Response · Authors · 2025-12-03
**Summary of Revisions**

Dear Area Chair and Reviewers,

We appreciate your stepping in to oversee our submission under the current circumstances. Below we provide a brief summary of how the discussion and revision have shaped the paper.

---

**General Overview.**

Across reviews, the paper was characterized as a **simple, practical mechanism** for **reducing the quadratic cost of visual world-model planning,** with **strong empirical support** in both simulation and real-robot settings. The revised manuscript is intended both to reinforce these acknowledged strengths and to directly address the major concerns raised during the discussion through a focused set of new experiments, analyses, and clarifications, all **highlighted in blue** in the updated manuscript.

---

**Reviewer-Recognized Strengths.**

We are encouraged that the reviewers found our proposed "Sparse Imagination" to be **well-motivated** (ASzo, gLWK), **clearly presented** (C7kW), and a **conceptually simple, practical, architecture-agnostic approach** that is broadly applicable to visual world-modeling settings (C7kW, HGmx, gLWK). Reviewers particularly highlighted the **counter-intuitive yet valuable insight** that random sampling consistently outperforms complex importance-based baselines (C7kW, HGmx) and the **interesting analysis** of token redundancy (ASzo). Furthermore, the **strong empirical performance** demonstrating significant planning speedups while maintaining success rates was widely acknowledged (ASzo, C7kW, gLWK), as was the **extensive validation** extending to real-world robotic control and VLA integration (C7kW, gLWK).

---

**Summary of Revisions and Responses**

**1. New Experiments & Empirical Results**

* **Expanded evaluation in real/simulated settings** (HGmx, C7kW; Sec. 1, 4.1, 4.3, App. B.1, B.8, D.1, D.2, Tables 5-6, Figs. 3, 5, 10, 14-16)

    Added a new real-robot task and Meta-World environment to demonstrate transferability to physical hardware and broader benchmarks while maintaining efficiency.


* **Additional planning paradigms** (ASzo, C7kW; App. C.5 & Table 8)

    Demonstrated effectiveness with an alternative planner (gradient-descent-based) to show the method is not tied to CEM-based MPC.


* **Broader encoder coverage** (ASzo, C7kW, HGmx; Sec. 5.1 & Fig. 19)

    Evaluated diverse visual backbones to confirm that the benefits of random token sampling are not tied to a specific encoder family.


* **Blind-spot behavior analysis** (HGmx; Sec. 5.4, Fig. 9, App. C.9 & Table 12, App. D.3 & Figs. 19-20)

    Provided visualizations and quantitative analysis confirming that random dropout reduces persistent blind spots compared to other strategies.


* **Exploring uncertainty-aware adaptive sparsity** (C7kW, gLWK; Sec. 6, App. C.6 & Table 9)

    Implemented a simple adaptive mechanism that dynamically adjusts sparsity based on model confidence, demonstrating support for uncertainty-aware compute adaptation.


* **Fine-grained compute/latency breakdown** (C7kW, gLWK; App. C.8 & Table 11)

    Decomposed end-to-end planning time to pinpoint latency sources.


* **World-model prediction quality** (ASzo; App. C.7 & Table 10)

    Verified that sparse training does not degrade full-patch prediction quality.


**2. Additional Discussion & Clarifications**

* **Clarifications and readability** (All Reviewers; Table 1, Table 2, Table 4, Sec. 4.1, 4.4, App. B.6)

    Clarified the research scope and the design rationale of grouped attention (emphasizing it is not purely empirical), alongside streamlined method descriptions and improved table readability.


* **Sensitivity to token-drop ratio** (C7kW, gLWK)

    Systematically reported performance across sparsity levels to identify the effective operating range.


* **Conceptual scope & novelty & characteristics** (ASzo, C7kW, gLWK)

    Refined discussions on operating conditions, novelty, and architectural characteristics.


* **Extended discussions** (HGmx, ASzo)

    Addressed non-goal-conditioned use, non-monotonic trends, and complementarity with other compression approaches (e.g., latent action models).


---

**Closing Remarks.**

We believe the new experimental results across **real-robot hardware, additional simulation benchmarks, and comparative studies**, **together with the clarified discussions**, **substantially strengthen the paper** and **effectively address the concerns raised**. We hope the revised manuscript clearly demonstrates the robustness and practicality of our approach. Thank you again for your time and effort in evaluating our work.

Best Regards,

Authors

---

### Meta-Review · Area_Chair_uuVd · 2026-01-02

**Summary:**

I suggest acceptance for this paper, informed by the following reviewer feedback:

**ASzo** states that the *strengths* are well-motivated for sparsity in WMs, the latency vs. performance tradeoff is very favorable, the analysis of token information in Section 5.3 is very interesting. The main *concerns* are (1) could the proposed approach be generalized beyond world models that use patch tokens?, (2) how are results affected since token sampling occurs only once during open loop planning, (3) need for more evaluation on the WM quality. These were all addressed appropriately in the rebuttal.

**C7kW** states that the *strengths* are the paper is well-written with a clear contribution, the core idea is simple but generally applicable, the method is appropriately compared, and the method is evaluated in real-world robotic control with a VLA. The main *concerns* are (1) the writing in the experiments section needs improvement, (2) it would be helpful to provide conceptual / theoretical justification for the random patch dropout robustness, (3) the need for more visual encoders and benchmark tasks, (4) evaluation of the approach with different styles of planners / planning paradigm, (5) finer-grained efficiency benchmarking on a per-component basis. These were all addressed appropriately in the rebuttal.

**HGmx** states that the *strengths* are the simple yet effective idea and extensive experiments.  The main *concerns* are (1) single task real-world eval limitation, (2) need for more visualizations and detailed comparisons. These were all addressed appropriately in the rebuttal.


**gLWK** states that the *strengths* are the simple yet effective idea and strong empirical validation. The main *concerns* are (1) no theoretical grounding, (2) need to study dropout ratio and other empirical design choices. These were all addressed appropriately in the rebuttal.

**Reviewer Concerns:**

**ASzo Concerns**
* (1) could the proposed approach be generalized beyond world models that use patch tokens? $\rightarrow$ **addressed (empirically and discussion)**
* (2) how are results affected since token sampling occurs only once during open loop planning $\rightarrow$ **partially addressed (discussion)**
* (3) need for more evaluation on the WM quality $\rightarrow$ **addressed (empirically)**

**C7kW Concerns**
* (1) the writing in the experiments section needs improvement  $\rightarrow$ **addressed (main text is edited significantly)**
* (2) it would be helpful to provide conceptual / theoretical justification for the random patch dropout robustness  $\rightarrow$ **partially addressed (theoretical justification is out of scope, but added more experimental structure to probe at this)**
* (3) the need for more visual encoders and benchmark tasks $\rightarrow$ **addressed (empirically)**
* (4) evaluation of the approach with different styles of planners / planning paradigm $\rightarrow$ **addressed (empirically, new MPC-GD planner added)**
* (5) finer-grained efficiency benchmarking on a per-component basis $\rightarrow$ **addressed (empirically)**

**HGmx Concerns**
* (1) single task real-world eval limitation $\rightarrow$  **partially addressed (working on a new task but time constraints and complexity prevent results from being reported now; reviewer still would like to see this)**
* (2) need for more visualizations and detailed comparisons. $\rightarrow$ **addressed (new figures in main text, more discussion back and forth with the reviewer)**

**gLWK Concerns**
* (1) no theoretical grounding  $\rightarrow$  **partially addressed (theoretical justification is out of scope, but added more experimental structure to probe at this)**
* (2) need to study dropout ratio and other empirical design choices  $\rightarrow$ **addressed (discussion)**

**Reviewer Scores:**

* **ASzo** would have *maintained* a score of 6: marginally above the acceptance threshold.
* **C7kW** would have *increased* to a score of 6: marginally above the acceptance threshold.
* **HGmx** would have *maintained* a score of 6: marginally above the acceptance threshold.
* **gLWK** would have *maintained* a score of 6: marginally above the acceptance threshold.

---

### Decision · Program_Chairs · 2026-01-26

Accept (Poster)